# LISFLOOD-FP 8.0: the new discontinuous Galerkin shallow water solver for multi-core CPUs and GPUs

James Shaw[1], Georges Kesserwani[1], Jeffrey Neal[2], Paul Bates[2], and Mohammad Kazem Sharifian[1]

[1]Department of Civil and Structural Engineering, The University of Sheffield, Western Bank, Sheffield, UK
[2]School of Geographical Sciences, University of Bristol, Bristol, UK

**Correspondence:** James Shaw (js102@zepler.net)

**Abstract.** LISFLOOD-FP 8.0 includes second-order discontinuous Galerkin (DG2) and first-order finite volume (FV1) solvers of the two-dimensional shallow water equations for modelling a wide range of flows, including rapidly-propagating, supercritical flows, shock waves, or flows over very smooth surfaces. The solvers are parallelised on multi-core CPU and Nvidia GPU architectures and run existing LISFLOOD-FP modelling scenarios without modification. These new, fully two-dimensional solvers are available alongside the existing local inertia solver (called ACC), which is optimised for multi-core CPUs and integrates with the LISFLOOD-FP sub-grid channel model. The predictive capabilities and computational scalability of the new DG2 and FV1 solvers are studied for two Environment Agency benchmark tests and a real-world fluvial flood simulation driven by rainfall across a 2500 $km^2$ catchment. DG2's second-order-accurate, piecewise-planar representation of topography and flow variables enables predictions on coarse grids that are competitive with FV1 and ACC predictions on 2–4$\times$ finer grids, particularly where river channels are wider than half the grid spacing. Despite the simplified formulation of the local inertia solver, ACC is shown to be spatially second-order-accurate and yields predictions that are close to DG2. The DG2-CPU and FV1-CPU solvers achieve near-optimal scalability up to 16 CPU cores and achieve greater efficiency on grids with fewer than 0.1 million elements. The DG2-GPU and FV1-GPU solvers are most efficient on grids with more than 1 million elements, where the GPU solvers are 2.5–4$\times$ faster than the corresponding 16-core CPU solvers. LISFLOOD-FP 8.0 therefore marks a new step towards operational DG2 flood inundation modelling at the catchment scale. LISFLOOD-FP 8.0 is freely available under the GPL v3 license, with additional documentation and case studies at https://www.seamlesswave.com/LISFLOOD8.0.

## 1 Introduction

LISFLOOD-FP is a freely-available raster-based hydrodynamic model that has been applied in numerous studies from small-scale (Sampson et al., 2012) and reach-scale (Liu et al., 2019; Shustikova et al., 2019; O'Loughlin et al., 2020) to continental and global flood forecasting applications (Wing et al., 2020; Sampson et al., 2015). LISFLOOD-FP has been coupled to several hydrological models (Hoch et al., 2019; Rajib et al., 2020; Li et al., 2020), and it offers simple text file configuration

and command-line tools to facilitate DEM preprocessing and sensitivity analyses (Sosa et al., 2020). LISFLOOD-FP includes extension modules to provide efficient rainfall routing (Sampson et al., 2013), modelling of hydraulic structures (Wing et al., 2019; Shustikova et al., 2020), and coupling between two-dimensional flood-plain solvers and a one-dimensional sub-grid channel model (Neal et al., 2012a).

LISFLOOD-FP already includes a local inertia (or 'gravity wave') solver, LISFLOOD-ACC, and a diffusive wave (or 'zero-inertia') solver, LISFLOOD-ATS. The LISFLOOD-ACC solver simplifies the full shallow water equations by neglecting convective acceleration, while LISFLOOD-ATS neglects both convective and inertial acceleration. The LISFLOOD-ACC solver is recommended for simulating fluvial, pluvial and coastal flooding, involving gradually-varying, subcritical flow over sufficiently rough surfaces with Manning's coefficient of at least $0.03 \text{ sm}^{-1/3}$ (Neal et al., 2012b; de Almeida and Bates, 2013). For such flows, LISFLOOD-ACC was reported to be up to $67\times$ faster than LISFLOOD-ATS, which has a stricter, quadratic CFL constraint (Neal et al., 2011; Hunter et al., 2006), and about $3\times$ faster than a full shallow water solver (Neal et al., 2012b). However, given the theoretical limitations of the local inertia equations (de Almeida and Bates, 2013; Martins et al., 2016; Cozzolino et al., 2019), a full shallow water solver is still required for simulating dam breaks (Neal et al., 2012b) and flash floods in steep catchments (Kvočka et al., 2017), involving rapidly-varying, supercritical flows, shock waves, or flows over very smooth surfaces.

The potential benefits of a second-order discontinuous Galerkin (DG2) shallow water solver for flood inundation modelling have recently been demonstrated by Ayog et al. (2021): DG2 alleviates numerical diffusion errors associated with first-order finite volume (FV1) methods, meaning DG2 can capture fine-scale transients in flood hydrographs on relatively coarse grids over long-duration simulations thanks to its piecewise-planar representation of topography and flow variables. Within a computational element on a raster grid, each locally-planar variable is represented by three coefficients—the element-average, $x$-slope and $y$-slope coefficients—which are updated by a two-stage Runge-Kutta time-stepping scheme. Due to its second-order formulation, DG2 can be $4$–$12\times$ slower per element than a FV1 solver depending on the test case (Kesserwani and Sharifian, 2020), though substantial speed-ups have already been achieved: switching from a standard tensor-product stencil to a simplified, slope-decoupled stencil of Kesserwani et al. (2018) achieved a $2.6\times$ speed-up, and avoiding unnecessary local slope limiting achieved an additional $2\times$ speed-up (Ayog et al., 2021), while preserving accuracy, conservation and robustness properties for shockless flows.

Second-order finite volume (FV2) methods offer an alternative approach to obtain second-order accuracy, with many FV2 models adopting the Monotonic Upstream-centred Scheme for Conservation Laws (MUSCL) method. While FV2-MUSCL solvers can achieve second-order convergence (Kesserwani and Wang, 2014), the MUSCL method relies on global slope limiting and non-local, linear reconstructions across neighbouring elements that can affect energy conservation properties (Ayog et al., 2021) and affect wave arrival times when the grid is too coarse (Kesserwani and Wang, 2014). Hence, although FV2-MUSCL is typically $2$–$10\times$ faster than DG2 per element (Ayog et al., 2021), DG2 can improve accuracy and conservation properties on coarse grids, which is particularly desirable for efficient, long-duration continental- or global-scale simulations that rely on DEM products derived from satellite data (Bates, 2012; Yamazaki et al., 2019).

Parallelisation is the next step towards making DG2 flood modelling operational on large-scale, high-resolution domains. Existing LISFLOOD-FP solvers are parallelised using OpenMP for multi-core CPUs, which have been tested on domains with up to 23 million elements on a 16-core CPU (Neal et al., 2009, 2018). But as flood models are applied to increasingly large

domains at increasingly fine resolutions, a greater degree of parallelism can be achieved using GPU accelerators (Brodtkorb et al., 2013). For example, García-Feal et al. (2018) compared Iber+ hydrodynamic model runs on a GPU against a 16-core CPU and obtained a 4–15× speed-up depending on the test case. Running in a multi-GPU configuration, the TRITON model has been applied on a 6800 $km^2$ domain with 68 million elements to simulate a 10-day storm event in under 30 minutes (Morales-Hernández et al., 2020b), and the HiPIMS model was applied on a 2500 $km^2$ domain with 100 million elements to

simulate a 4-day storm event in 1.5 days (Xia et al., 2019).

This paper presents a new LISFLOOD-DG2 solver of the full shallow water equations, which is integrated into LISFLOOD-FP 8.0 and freely available under the GNU GPL v3 license (LISFLOOD-FP developers, 2020). LISFLOOD-FP 8.0 also includes an updated FV1 solver obtained by simplifying the DG2 formulation. Both solvers support standard LISFLOOD-FP configuration parameters and model outputs, meaning that many existing LISFLOOD-FP modelling scenarios can run without

modification. Since the new DG2 and FV1 solvers are purely two-dimensional and parallelised for multi-core CPU and GPU architectures, the new solvers do not currently integrate with the LISFLOOD-FP sub-grid channel model (Neal et al., 2012a) or incorporate the CPU-specific optimisations available to the ACC solver (Neal et al., 2018).

The paper is structured as follows: Sect. 2 presents the LISFLOOD-DG2 and FV1 formulations, and the parallelisation strategies using OpenMP for multi-core CPU architectures and CUDA for Nvidia GPU architectures. Sect. 3 evaluates the

DG2, FV1 and ACC solvers across three flood inundation test cases. The first two cases reproduce Environment Agency benchmark tests (Néelz and Pender, 2013): the first case simulates a slowly-propagating wave over a flat floodplain, measuring computational scalability on multi-core CPU and GPU architectures and comparing the spatial grid convergence of DG2, FV1 and ACC predictions; the second case simulates a rapidly-propagating wave along a narrow valley with irregular topography, assessing the solver capabilities for modelling supercritical flow. The final case reproduces fluvial flooding over the 2500 $km^2$

Eden catchment in North West England, caused by Storm Desmond in December 2015 (Xia et al., 2019). This is the first assessment of a DG2 hydrodynamic model in simulating a real-world storm event at catchment scale, with overland flow driven entirely by spatially- and temporally-varying rainfall data. Concluding remarks are made in Sect. 4. Additional LISFLOOD-FP 8.0 documentation and further test cases are available at https://www.seamlesswave.com/LISFLOOD8.0.

## 2   The LISFLOOD-FP model

LISFLOOD-FP 8.0 includes a new second-order discontinuous Galerkin (DG2) solver and an updated first-order finite volume (FV1) solver that simulate two-dimensional shallow water flows. The new DG2 and FV1 formulations and the existing LISFLOOD-ACC formulation are described in the following subsections.

## 2.1 The new LISFLOOD-DG2 solver

The LISFLOOD-DG2 solver implements the DG2 formulation of Kesserwani et al. (2018) that adopts a simplified 'slope-decoupled' stencil compatible with raster-based Godunov-type finite volume solvers. Piecewise-planar topography, water depth and discharge fields are modelled by an element-average coefficient and dimensionally-independent $x$-slope and $y$-slope coefficients. This DG2 formulation achieves well-balancedness for all discharge coefficients in the presence of irregular, piecewise-planar topography with wetting-and-drying (Kesserwani et al., 2018). A piecewise-planar treatment of the friction term is applied to all discharge coefficients prior to each time-step, based on the split implicit friction scheme of Liang and Marche (2009). Informed by the findings of Ayog et al. (2021), the automatic local slope limiter option in LISFLOOD-DG2 is deactivated for the flood-like test cases presented in Sect. 3. This slope-decoupled, no-limiter approach can achieve a $5\times$ speed-up over a standard tensor-product stencil with local slope limiting (Kesserwani et al., 2018; Ayog et al., 2021), meaning this DG2 formulation is expected to be particularly efficient for flood modelling applications.

The DG2 formulation discretises the two-dimensional shallow water equations, written in conservative vectorial form as

$$\partial_t \mathbf{U} + \partial_x \mathbf{F}(\mathbf{U}) + \partial_y \mathbf{G}(\mathbf{U}) = \mathbf{S}_b(\mathbf{U}) + \mathbf{S}_f(\mathbf{U}) + \mathbf{R}, \tag{1}$$

where $\partial_t$, $\partial_x$ and $\partial_t$ denote partial derivatives in the horizontal spatial dimensions $x$ and $y$, and temporal dimension $t$. In Eqn. (1), $\mathbf{U}$ is the vector of flow variables, $\mathbf{F}(\mathbf{U})$ and $\mathbf{G}(\mathbf{U})$ are flux vectors in the $x$- and $y$-directions, and $\mathbf{S}_b$, $\mathbf{S}_f$ and $\mathbf{R}$ are source terms representing the topographic slope, frictional force, and rainfall:

$$\mathbf{U} = \begin{bmatrix} h \\ q_x \\ q_y \end{bmatrix}, \ \mathbf{F} = \begin{bmatrix} q_x \\ \frac{q_x^2}{h} + \frac{g}{2}h^2 \\ \frac{q_x q_y}{h} \end{bmatrix}, \ \mathbf{G} = \begin{bmatrix} q_y \\ \frac{q_x q_y}{h} \\ \frac{q_y^2}{h} + \frac{g}{2}h^2 \end{bmatrix},$$

$$\mathbf{S}_b = \begin{bmatrix} 0 \\ -gh\partial_x z \\ -gh\partial_y z \end{bmatrix}, \ \mathbf{S}_f = \begin{bmatrix} 0 \\ S_{fx} \\ S_{fy} \end{bmatrix}, \ \mathbf{R} = \begin{bmatrix} R \\ 0 \\ 0 \end{bmatrix}, \tag{2}$$

with water depth $h$ [L], unit-width discharges $q_x = hu$ and $q_y = hv$ [L$^3$/T], and depth-averaged horizontal velocities $u$ and $v$ [L/T] in the $x$- and $y$-directions respectively. Units are notated in square brackets [·], where L denotes unit length and T denotes unit time. The two-dimensional topographic elevation data is denoted $z$ [L] and $g$ is the gravitational acceleration [L/T$^2$]. The frictional forces in the $x$- and $y$-directions are $S_{fx} = -C_f u\sqrt{u^2 + v^2}$ and $S_{fy} = -C_f v\sqrt{u^2 + v^2}$, where the friction function is $C_f = gn_M^2/h^{1/3}$ and $n_M(x,y)$ is Manning's coefficient [T/L$^{1/3}$]. The prescribed rainfall rate is given by $R(x,y,t)$ [L/T].

The DG2 discretisation of Eqn. (1) is compatible with existing LISFLOOD-FP data structures, being formulated on a raster grid of uniform rectangular elements. A rectangular element is shown in Fig. 1, centred at $(x_{i,j}, y_{i,j})$ with horizontal dimensions $(\Delta x, \Delta y)$. Within the element the discrete flow vector $\mathbf{U_h}(x,y)$ and topography $z_\mathbf{h}(x,y)$ are represented by locally-planar fields. Expressed as a scaled Legendre basis expansion (Kesserwani and Sharifian, 2020), the flow vector $\mathbf{U_h}(x,y)$ is written

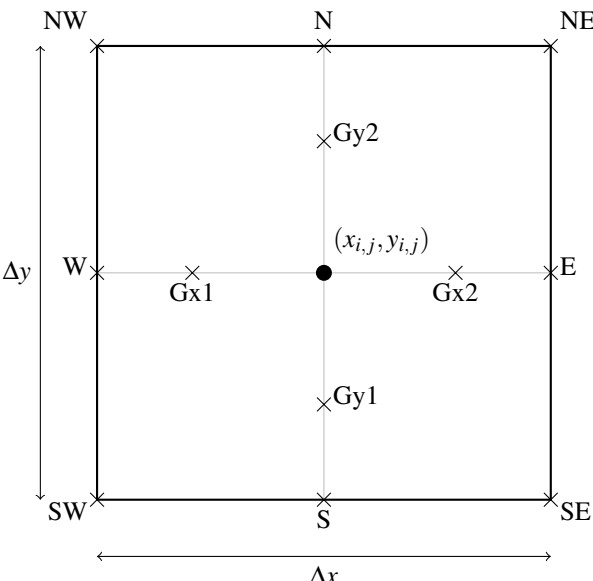

**Figure 1.** DG2 slope-decoupled stencil defined on a rectangular element centred at $(x_{i,j}, y_{i,j})$ with horizontal dimensions $(\Delta x, \Delta y)$. N, E, S, W mark the northern, eastern, southern and western face centres, and Gx1, Gx2, Gy1 and Gy2 mark the four Gaussian quadrature points.

as:

$$\mathbf{U_h}(x,y) = \mathbf{U}_{i,j} \begin{bmatrix} 1 \\ 2\sqrt{3}\,(x - x_{i,j})/\Delta x \\ 2\sqrt{3}\,(y - y_{i,j})/\Delta y \end{bmatrix}, \tag{3}$$

where $\mathbf{U}_{i,j}$ is the matrix of flow coefficients:

$$\mathbf{U}_{i,j} = \begin{bmatrix} h_{i,j,0} & h_{i,j,1x} & h_{i,j,1y} \\ q_{x_{i,j,0}} & q_{x_{i,j,1x}} & q_{x_{i,j,1y}} \\ q_{y_{i,j,0}} & q_{y_{i,j,1x}} & q_{y_{i,j,1y}} \end{bmatrix}, \tag{4}$$

in which subscript $0$ denotes the element-average coefficients, and subscript $1x$ and $1y$ denote the linear slope coefficients in the $x$- and $y$-directions. The topography coefficients are:

$$z_{i,j} = [z_{i,j,0}, z_{i,j,1x}, z_{i,j,1y}], \tag{5}$$

which are initialised from a DEM raster file as described later in Sect. 2.1.1. Assembling all elements onto a raster grid yields piecewise-planar representations of topography and flow variables that intrinsically capture smooth, linear variations within
each element, while simultaneously allowing flow discontinuities—such as hydraulic jumps and shock waves—to be captured at element interfaces.

By adopting the slope-decoupled form of Kesserwani et al. (2018) that uses the local stencil shown in Fig. 1, the locally-planar solution is easily evaluated at the four face centres (denoted N, S, E, W):

$$\mathbf{U}_{i,j}^{\mathrm{W}} = \mathbf{U}_{i,j,0} - \sqrt{3}\mathbf{U}_{i,j,1x}, \; \mathbf{U}_{i,j}^{\mathrm{E}} = \mathbf{U}_{i,j,0} + \sqrt{3}\mathbf{U}_{i,j,1x},$$

$$\mathbf{U}_{i,j}^{\mathrm{S}} = \mathbf{U}_{i,j,0} - \sqrt{3}\mathbf{U}_{i,j,1y}, \; \mathbf{U}_{i,j}^{\mathrm{N}} = \mathbf{U}_{i,j,0} + \sqrt{3}\mathbf{U}_{i,j,1y}, \tag{6}$$

and at the four Gaussian quadrature points (denoted Gx1, Gx2, Gy1 and Gy2):

$$\mathbf{U}_{i,j}^{\mathrm{Gx1}} = \mathbf{U}_{i,j,0} - \mathbf{U}_{i,j,1x}, \; \mathbf{U}_{i,j}^{\mathrm{Gx2}} = \mathbf{U}_{i,j,0} + \mathbf{U}_{i,j,1x},$$

$$\mathbf{U}_{i,j}^{\mathrm{Gy1}} = \mathbf{U}_{i,j,0} - \mathbf{U}_{i,j,1y}, \; \mathbf{U}_{i,j}^{\mathrm{Gy2}} = \mathbf{U}_{i,j,0} + \mathbf{U}_{i,j,1y}. \tag{7}$$

A standard splitting approach is adopted such that the friction source term $\mathbf{S}_f$ and rainfall source term $\mathbf{R}$ in Eqn. (1) are applied separately at the beginning of each time-step. By adopting a splitting approach, friction or rainfall source terms are only applied as required by the particular test case, for better runtime efficiency. The discretisation of the friction source term is described later in Sect. 2.1.2, and the rainfall source term in Sect. 2.1.3. The remaining terms are the spatial fluxes and topographic slope terms, which are discretised by an explicit second-order two-stage Runge-Kutta scheme (Kesserwani et al., 2010) to evolve the flow coefficients $\mathbf{U}_{i,j}$ from time level $n$ to $n+1$:

$$\mathbf{U}^{\mathrm{int}} = \mathbf{U}^n + \Delta t \, \mathbf{L}(\mathbf{U}^n), \tag{8a}$$

$$\mathbf{U}^{n+1} = \frac{1}{2}\left[\mathbf{U}^n + \mathbf{U}^{\mathrm{int}} + \Delta t \mathbf{L}(\mathbf{U}^{\mathrm{int}})\right], \tag{8b}$$

where element indices $(i,j)$ are omitted for clarity of presentation. The initial time-step $\Delta t$ is a fixed value specified by the user, and the time-step is updated thereafter according to the CFL condition using the maximum stable Courant number of 0.33 (Cockburn and Shu, 2001). The spatial operator $\mathbf{L} = [\mathbf{L}_0, \mathbf{L}_{1x}, \mathbf{L}_{1y}]$ is:

$$\mathbf{L}_0(\mathbf{U}_{i,j}) =$$

$$-\left(\frac{\widetilde{\mathbf{F}}_{\mathrm{E}} - \widetilde{\mathbf{F}}_{\mathrm{W}}}{\Delta x} + \frac{\widetilde{\mathbf{G}}_{\mathrm{N}} - \widetilde{\mathbf{G}}_{\mathrm{S}}}{\Delta y} + \begin{bmatrix} 0 \\ 2\sqrt{3}g\overline{h}_{i,j,0x}\overline{z}_{i,j,1x}/\Delta x \\ 2\sqrt{3}g\overline{h}_{i,j,0y}\overline{z}_{i,j,1y}/\Delta y \end{bmatrix}\right), \tag{9a}$$

$$\mathbf{L}_{1x}(\mathbf{U}_{i,j}) =$$

$$-\frac{\sqrt{3}}{\Delta x}\left(\widetilde{\mathbf{F}}_{\mathrm{W}} + \widetilde{\mathbf{F}}_{\mathrm{E}} - \mathbf{F}(\overline{\mathbf{U}}_{i,j}^{\mathrm{Gx1}}) - \mathbf{F}(\overline{\mathbf{U}}_{i,j}^{\mathrm{Gx2}}) + \begin{bmatrix} 0 \\ 2g\overline{h}_{i,j,1x}\overline{z}_{i,j,1x} \\ 0 \end{bmatrix}\right), \tag{9b}$$

$$\mathbf{L}_{1y}(\mathbf{U}_{i,j}) =$$

$$-\frac{\sqrt{3}}{\Delta y}\left(\widetilde{\mathbf{G}}_{\mathrm{S}} + \widetilde{\mathbf{G}}_{\mathrm{N}} - \mathbf{G}(\overline{\mathbf{U}}_{i,j}^{\mathrm{Gy1}}) - \mathbf{G}(\overline{\mathbf{U}}_{i,j}^{\mathrm{Gy2}}) + \begin{bmatrix} 0 \\ 0 \\ 2g\overline{h}_{i,j,1y}\overline{z}_{i,j,1y} \end{bmatrix}\right), \tag{9c}$$

in which variables with an overline denote temporary modifications to the original variables that ensure well-balancedness and non-negative water depths (Kesserwani et al., 2018; Liang and Marche, 2009), and $\widetilde{\mathbf{F}}_\text{W}$, $\widetilde{\mathbf{F}}_\text{E}$, $\widetilde{\mathbf{G}}_\text{S}$, $\widetilde{\mathbf{G}}_\text{N}$ denote HLL approximate Riemann fluxes across western, eastern, northern and southern interfaces. Each Riemann solution resolves the discontinuity between the flow variables evaluated at the limits of the locally-planar solutions adjacent to the interface. Because of the locally-planar nature of the DG2 solutions, such a discontinuity is likely to be very small when the flow is smooth—as is often the case for flood inundation events—and won't be significantly enlarged by grid coarsening.

While LISFLOOD-DG2 is equipped with a generalised minmod slope limiter (Cockburn and Shu, 2001) localised by the Krivodonova shock detector (Krivodonova et al., 2004), the automatic local slope limiter was deactivated for the sake of efficiency: none of the test cases presented in Sect. 3 involve shock wave propagation since all waves propagate over an initially dry bed and are rapidly retarded by frictional forces (Néelz and Pender, 2013; Xia et al., 2019). The lack of shock wave propagation means that all LISFLOOD-FP solvers—DG2, FV1 and ACC—are capable of realistically simulating all test cases presented in Sect. 3.

### 2.1.1 Initialisation of piecewise-planar topography coefficients from a DEM raster file

The topography coefficients $[z_{i,j,0}, z_{i,j,1x}, z_{i,j,1y}]$ are initialised to ensure the resulting piecewise-planar topography is continuous at face centres, where Riemann fluxes are calculated and the wetting-and-drying treatment is applied under the well-balancedness property (Kesserwani et al., 2018). The topographic elevations at the N, S, E, and W face centres are calculated by averaging the DEM raster values taken at the NW, NE, SW and SE vertices (Fig. 1) such that $z_{i,j}^\text{N} = (z_{i,j}^\text{NW} + z_{i,j}^\text{NE})/2$ and similarly for $z_{i,j,}^\text{E}$, $z_{i,j,}^\text{S}$, and $z_{i,j,}^\text{W}$. The element-average coefficient $z_{i,j,0}$ is then calculated as:

$$z_{i,j,0} = \frac{1}{4}\left[ z_{i,j}^\text{NW} + z_{i,j}^\text{SW} + z_{i,j}^\text{NE} + z_{i,j}^\text{SE} \right], \tag{10a}$$

while the slope coefficients $z_{i,j,1x}$ and $z_{i,j,1y}$ are calculated as the gradients across opposing face centres:

$$z_{i,j,1x} = \frac{1}{2\sqrt{3}}\left( z_{i,j}^\text{E} - z_{i,j}^\text{W} \right), \tag{10b}$$

$$z_{i,j,1y} = \frac{1}{2\sqrt{3}}\left( z_{i,j}^\text{N} - z_{i,j}^\text{S} \right). \tag{10c}$$

LISFLOOD-FP 8.0 includes a utility application, `generateDG2DEM`, that loads an existing DEM raster file and outputs new raster files containing the element-average, $x$-slope and $y$-slope topography coefficients, ready to be loaded by the LISFLOOD-DG2 solver.

### 2.1.2 Discretisation of the friction source term

The discretisation of the friction source term is based on the split implicit scheme of Liang and Marche (2009). Without numerical stabilisation, the friction function $C_f = gn_M^2/h^{1/3}$ can grow exponentially as the water depth vanishes at a wet-dry front, but the scheme adopted here is designed to ensure numerical stability by limiting the frictional force to prevent unphysical flow reversal.

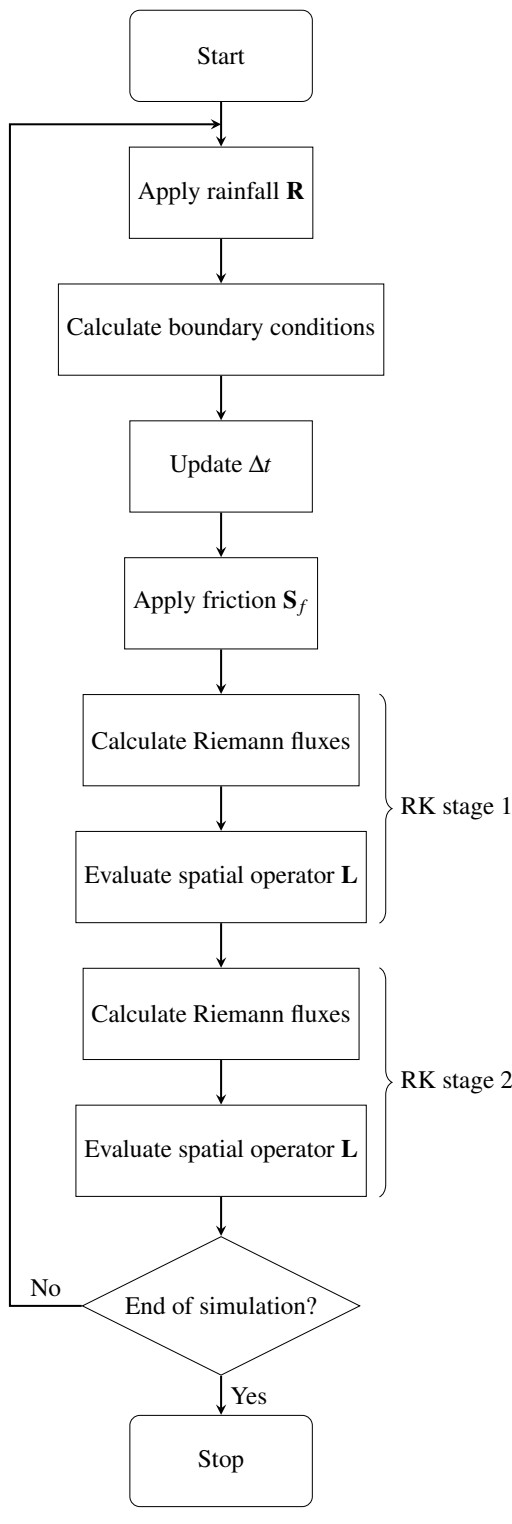

**Figure 2.** Flowchart of operations for the DG2 formulation (Sect. 2.1).

The implicit friction scheme is solved directly (see Liang and Marche, 2009, Sect. 3.4) such that frictional forces are applied to the $x$-directional discharge component $q_x$ over a time-step $\Delta t$, yielding a retarded discharge component $q_{fx}$:

$$q_{fx}(\mathbf{U}) = q_x + \Delta t \frac{S_{fx}}{\mathcal{D}_x}, \tag{11a}$$

where the denominator $\mathcal{D}_x$ is

$$\mathcal{D}_x = 1 + \left(\frac{\Delta t C_f}{h}\right)\left(\frac{2u^2 + v^2}{\sqrt{u^2 + v^2}}\right). \tag{11b}$$

To update the element-average discharge coefficient $q_{x\,i,j,0}$, Eqn. (11) is evaluated at the element centre :

$$q_{x\,i,j,0}^{n+1} = q_{fx}(\mathbf{U}_{i,j,0}^n), \tag{12a}$$

while the slope coefficients $q_{x\,i,j,1x}$ and $q_{x\,i,j,1y}$ are updated by calculating the $x$- and $y$-gradients using evaluations of Eqn. (11) at Gaussian quadrature points Gx1, Gx2, and Gy1, Gy2 (Fig. 1):

$$q_{x\,i,j,1x}^{n+1} = \frac{1}{2}\left[q_{fx}(\mathbf{U}_{i,j}^{\mathrm{Gx2}}) - q_{fx}(\mathbf{U}_{i,j}^{\mathrm{Gx1}})\right], \tag{12b}$$

$$q_{x\,i,j,1y}^{n+1} = \frac{1}{2}\left[q_{fx}(\mathbf{U}_{i,j}^{\mathrm{Gy2}}) - q_{fx}(\mathbf{U}_{i,j}^{\mathrm{Gy1}})\right]. \tag{12c}$$

Similarly, frictional forces are applied to the $y$-directional discharge component $q_y$ yielding a retarded discharge $q_{fy}$:

$$q_{fy}(\mathbf{U}) = q_y + \Delta t \frac{S_{fy}}{\mathcal{D}_y}, \tag{13a}$$

$$\mathcal{D}_y = 1 + \left(\frac{\Delta t C_f}{h}\right)\left(\frac{u^2 + 2v^2}{\sqrt{u^2 + v^2}}\right). \tag{13b}$$

While this friction scheme has been successfully adopted in finite-volume and discontinuous Galerkin settings for modelling dam break flows and urban flood events (Wang et al., 2011; Kesserwani and Wang, 2014), it can exhibit spuriously large velocities and correspondingly small time-steps for large-scale, rainfall-induced overland flows, involving widespread, very thin water layers flowing down hill slopes and over steep river banks, as demonstrated by Xia et al. (2017). Due to the involvement of the slope coefficients, water depths at Gaussian quadrature points can be much smaller (and velocities much larger) than the element-average values. Therefore, for overland flow simulations, the LISFLOOD-DG2 time-step size is expected to be substantially reduced compared to LISFLOOD-FV1, which only involves element-average values.

### 2.1.3 Discretisation of the rainfall source term

The discretisation of rainfall source term evolves the water depth element-average coefficients $h_{i,j,0}$:

$$h_{i,j,0}^{n+1} = h_{i,j,0}^n + \Delta t R_{i,j}^n, \tag{14}$$

where $R_{i,j}^n$ denotes the prescribed rainfall rate at element $(i,j)$ and time level $n$. Eqn. (14) is first-order-accurate in space and time, which is deemed sufficient since rainfall data is typically available at far coarser spatial and temporal resolutions than the computation grid, leading to zero element-wise slope coefficients for the rainfall source term.

(a) OpenMP

```
#pragma omp parallel for
for (int j=0; j<Ny; j++) {
  for (int i=0; i<Nx; i++) {
    apply_update_operation();
  }
}
```

(b) CUDA

```
int global_i =
  blockIdx.x*blockDim.x + threadIdx.x;
int global_j =
  blockIdx.y*blockDim.y + threadIdx.y;

for (int j=global_j; j<Ny;
    j+=blockDim.y*gridDim.y) {
  for (int i=global_i; i<Nx;
      i+=blockDim.x*gridDim.x) {
    apply_update_operation();
  }
}
```

**Figure 3.** (a) OpenMP nested loop implementation to apply any update operation across a grid of (Nx × Ny) elements, processing rows in parallel; (b) CUDA nested grid-stride loop implementation to process 2D blocks in parallel.

Recall that the rainfall source term, friction source term, and remaining flux and bed slope terms are treated separately such that, at each timestep, the flow variables updated by Eqn. (14) are subsequently updated by Eqn. (12), and finally by Eqns. (8)–(9). The complete DG2 model workflow is summarised by the flowchart in Fig. 2, wherein each operation is parallelised using the CPU and GPU parallelisation strategies discussed next.

### 2.1.4 OpenMP parallelisation for multi-core CPUs

The LISFLOOD-DG2-CPU solver adopts OpenMP to process rows of the computational grid in parallel using the nested loop structure in Fig. 3a, which is applied to each operation in the flowchart in Fig. 2. The global time-step $\Delta t$ is found by calculating the minimum value across all elements using an OpenMP reduction. The same parallelisation strategy is already adopted in existing LISFLOOD-FP solvers (Neal et al., 2009) because it is straightforward to implement with minimal code changes for any explicit numerical scheme involving local, element-wise operations. While some LISFLOOD-FP solvers implement

more sophisticated OpenMP parallelisation and dry cell optimisation (Neal et al., 2018), this can introduce additional code complexity and runtime overhead (Morales-Hernández et al., 2020a), so it has not been adopted for the new LISFLOOD-DG2-CPU solver.

### 2.1.5 CUDA parallelisation for Nvidia GPUs

The LISFLOOD-DG2-GPU solver adopts a different parallelisation strategy using nested CUDA grid-stride loops (Fig. 3b), which is a recommended technique for parallel processing of raster data on GPUs (Harris, 2013). Using this strategy, a $16\times16$-element region of the computational grid is mapped to a CUDA block of $16\times16$ threads. Threads within each block execute in parallel, and multiple blocks also execute in parallel, thanks to the two-layer parallelism in the CUDA programming model. Nested grid-stride loops are applied to each operation in Fig. 2. Thanks to the localisation of DG2, almost all operations are evaluated element-wise and only require data available locally within the element. The only non-local operations are: (i) the global time-step, which is calculated using a min() reduction operator from the CUB library (Merrill, 2015), and (ii) the Riemann fluxes that connect flow discontinuities across interfaces between neighbouring elements, which are discussed next.

To process Riemann fluxes efficiently, the LISFLOOD-DG2-GPU solver adopts a new dimensionally-split form that allows expensive Riemann flux evaluations to be stored temporarily in low-latency shared memory on the GPU device (Qin et al., 2019). The new dimensionally-split form is derived by decomposing the spatial operator (Eqn. (9)) and the two-stage Runge-Kutta scheme (Eqn. (8)) into separate $x$- and $y$-directional updates. The slope-decoupled form allows a straightforward splitting of the spatial operator $\mathbf{L}$ in Eqn. (9) into an $x$-directional operator $\mathbf{L}_x = [\mathbf{L}_{0x}, \mathbf{L}_{1x}, \mathbf{0}]$ and a $y$-directional operator $\mathbf{L}_y = [\mathbf{L}_{0y}, \mathbf{0}, \mathbf{L}_{1y}]$ such that $\mathbf{L} = \mathbf{L}_x + \mathbf{L}_y$. The $\mathbf{L}_{1x}$ and $\mathbf{L}_{1y}$ operators are given in Eqn. (9b) and (9c), and $\mathbf{L}_{0x}$ and $\mathbf{L}_{0y}$ are defined as:

$$\mathbf{L}_{0x}(\mathbf{U}_{i,j}^n) = -\left( \frac{\widetilde{\mathbf{F}}_{\mathrm{E}} - \widetilde{\mathbf{F}}_{\mathrm{W}}}{\Delta x} + \begin{bmatrix} 0 \\ 2\sqrt{3}g\overline{h}_{i,j,0x}\overline{z}_{i,j,1x}/\Delta x \\ 0 \end{bmatrix} \right), \tag{15}$$

$$\mathbf{L}_{0y}(\mathbf{U}_{i,j}^n) = -\left( \frac{\widetilde{\mathbf{G}}_{\mathrm{N}} - \widetilde{\mathbf{G}}_{\mathrm{S}}}{\Delta y} + \begin{bmatrix} 0 \\ 0 \\ 2\sqrt{3}g\overline{h}_{i,j,0y}\overline{z}_{i,j,1y}/\Delta y \end{bmatrix} \right). \tag{16}$$

Similarly, each of the two Runge-Kutta stages in Eqn. (8) is split into two substages: the first updates the flow in the $x$-direction by applying $\mathbf{L}_x$; the second updates the flow in the $y$-direction by applying $\mathbf{L}_y$:

$$\mathbf{U}^{\mathrm{int},x} = \mathbf{U}^n + \Delta t\,\mathbf{L}_x(\mathbf{U}^n), \tag{17a}$$

$$\mathbf{U}^{\mathrm{int}} = \mathbf{U}^{\mathrm{int},x} + \Delta t\,\mathbf{L}_y(\mathbf{U}^n), \tag{17b}$$

$$\mathbf{U}^{n+1,x} = \frac{1}{2}\left[\mathbf{U}^n + \mathbf{U}^{\mathrm{int}} + \Delta t\mathbf{L}_x(\mathbf{U}^{\mathrm{int}})\right], \tag{17c}$$

$$\mathbf{U}^{n+1} = \mathbf{U}^{n+1,x} + \frac{1}{2}\Delta t\mathbf{L}_y(\mathbf{U}^{\mathrm{int}}). \tag{17d}$$

Each substage of Eqn. (17) is evaluated element-wise within a nested grid-stride loop. Within the $x$-directional spatial operator $\mathbf{L}_x$, the $x$-directional Riemann fluxes, $\widetilde{\mathbf{F}}_{\mathrm{E}}$ and $\widetilde{\mathbf{F}}_{\mathrm{W}}$, are calculated as follows:

1. thread $(i,j)$ calculates the Riemann flux across the eastern face of element $(i,j)$, $\widetilde{\mathbf{F}}_{\mathrm{E}}$, storing the result in a local variable, and in a shared memory array;

2. a synchronisation barrier waits for all threads in the CUDA block to complete;

3. thread $(i,j)$ then loads $\widetilde{\mathbf{F}}_{\mathrm{W}}$ from shared memory, which is the same as $\widetilde{\mathbf{F}}_{\mathrm{E}}$ already calculated by thread $(i-1,j)$ and,

4. finally, with $\widetilde{\mathbf{F}}_{\mathrm{E}}$ already stored as a local variable and $\widetilde{\mathbf{F}}_{\mathrm{W}}$ loaded from shared memory, thread $(i,j)$ can evaluate the $x$-direction operator $\mathbf{L}_x$.

The $y$-directional Riemann fluxes $\widetilde{\mathbf{G}}_{\mathrm{S}}$ and $\widetilde{\mathbf{G}}_{\mathrm{N}}$, within the $y$-directional operator $\mathbf{L}_y$ are calculated in the same way. By caching flux evaluations in low-latency shared memory, this dimensionally-split approach minimises the number of expensive Riemann flux evaluations and only requires a single synchronisation barrier within each CUDA block.

## 2.2 The new FV1 solver

While LISFLOOD-FP already includes a first-order finite volume solver called LISFLOOD-Roe (Villanueva and Wright, 2006; Neal et al., 2012b), LISFLOOD-FP 8.0 includes an updated FV1 solver that is parallelised for multi-core CPU and GPU architectures. The new FV1 formulation is obtained by simplifying the DG2 formulation (Sect. 2.1) to remove the slope coefficients and associated $\mathbf{L}_{1x}$ and $\mathbf{L}_{1y}$ spatial operators, yielding piecewise-constant representations of topography and flow variables. Like DG2, flow discontinuities at element interfaces are captured by FV1's piecewise-constant representation but, unlike DG2, smooth solutions cannot be captured without introducing artificial discontinuities, due to the lack of slope information within each element. Hence, FV1 is more vulnerable to grid coarsening since artificial discontinuities between elements tend to be enlarged as the grid becomes coarser, leading to increased numerical diffusion errors.

The LISFLOOD-FV1 formulation uses a standard first-order forward Euler time-stepping scheme (Eqn. (8a) with $\mathbf{U}^{n+1} = \mathbf{U}^{\mathrm{int}}$). The well-balanced wetting-and-drying treatment necessitates a maximum stable Courant number of 0.5 (Kesserwani and Liang, 2012).

## 2.3 The existing LISFLOOD-ACC local inertia solver

The LISFLOOD-ACC solver (Bates et al., 2010) adopts a hybrid finite-volume/finite-difference discretisation of the local inertia equations, which simplify the full shallow water equations by neglecting convective acceleration. Like LISFLOOD-FV1, LISFLOOD-ACC adopts the finite volume method to provide a piecewise-constant representation of water depth, evolved element-wise via the discrete mass conservation equation:

$$
h_{i,j}^{n+1} = h_{i,j}^n +
$$
$$
\frac{\Delta t}{\Delta x}\left(q_{x\,i-1/2,j}^{n+1} - q_{x\,i+1/2,j}^{n+1} + q_{y\,i,j-1/2}^{n+1} - q_{y\,i,j+1/2}^{n+1}\right), \tag{18}
$$

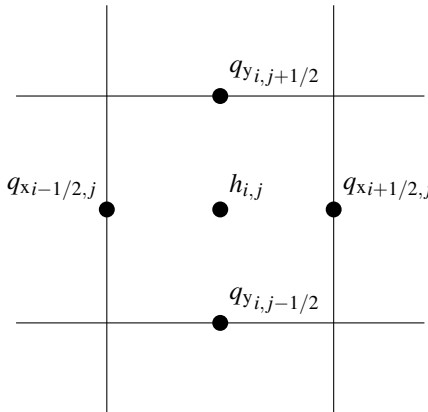

**Figure 4.** Staggered-grid arrangement of variables in the LISFLOOD-ACC formulation. Continuous discharge components $q_x$ and $q_y$ are stored normal to the face, and water depth $h$ is represented as a locally-constant value, stored at the element centre.

where the time-step $\Delta t$ is calculated using the default Courant number of 0.7.

Unlike LISFLOOD-FV1, LISFLOOD-ACC adopts a finite difference method to simplify the representation of inter-elemental fluxes by storing a single, continuous discharge component at each interface, leading to the so-called Arakawa C-grid staggering (Arakawa and Lamb, 1977) shown in Fig. 4. The discharge components are evolved via a simplified form of the momentum conservation equation coupled to the Manning friction formula: the $q_x$ discharge component at interface $(i-1/2, j)$ is evolved as: (Bates et al., 2010; de Almeida et al., 2012)

$$q_{x\,i-1/2,j}^{n+1} = \frac{q_{x\,i-1/2,j}^{n} - g\,h_f \frac{\Delta t}{\Delta x}\left(\eta_{i,j}^n - \eta_{i-1,j}^n\right)}{1 + g\,\Delta t\,n_M^2\left|q_{x\,i-1/2,j}^n/h_f^{7/3}\right|}, \tag{19}$$

where the numerical flow depth at the interface is $h_f = \max(\eta_{i,j}^n, \eta_{i-1,j}^n) - \max(z_{i,j}, z_{i-1,j})$. The $q_y$ discharge component is evolved in the same way.

As seen in Eqn. 19, the evolution of the continuous $q_x$ value at the interface only relies on a local reconstruction of the water surface gradient, $(\eta_{i,j}^n - \eta_{i-1,j}^n)/\Delta x$. This formulation could make LISFLOOD-ACC less sensitive than LISFLOOD-FV1 to grid coarsening for modelling flood inundation events, when the water surface varies smoothly. The Arakawa C-grid staggering adopted by LISFLOOD-ACC is commonly used in numerical weather prediction models (Collins et al., 2013) because it yields second-order-accuracy in space on a compact, local stencil. The second-order spatial accuracy of LISFLOOD-ACC is confirmed based on the numerical analysis of de Almeida et al. (2012), as presented in Appendix B.

## 3  Numerical results

Three simulations are performed to assess the computational scalability and predictive capability of LISFLOOD-DG2 compared with LISFLOOD-FV1 and LISFLOOD-ACC. The optimised LISFLOOD-ACC solver specified by Neal et al. (2018) implements a sub-grid channel model (Neal et al., 2012a) and CPU-specific optimisations that do not translate naturally to

GPU architectures. Additionally, at the time that model runs were performed, the optimised ACC solver did not yet support the rain-on-grid features used later in Sect. 3.3[1]. To facilitate a like-for-like intercomparison between solvers, the LISFLOOD-

ACC solver used here is the version specified by Neal et al. (2012b), which already supports the necessary rain-on-grid features and shares the same algorithmic approach as the FV1 and DG2 solvers.

The CPU solvers were run on a 2GHz Intel Xeon Gold 6138 using up to 16 CPU cores (with hyperthreading disabled), which is the maximum number of cores used in the LISFLOOD-FP parallelisation study of Neal et al. (2018). The GPU solvers were run on an Nvidia Tesla V100. LISFLOOD-FP is configured with double precision for all calculations. Simulation results are

openly available on Zenodo (Shaw et al., 2021).

### 3.1  Slowly-propagating wave over a flat floodplain

This synthetic test, known as Test 4 in Néelz and Pender (2013), is widely used to assess flood model predictions of slowly-propagating flow over a flat floodplain with high roughness (Neal et al., 2012b; Jamieson et al., 2012; Martins et al., 2015; Guidolin et al., 2016; Huxley et al., 2017). Since the floodplain is flat, the test setup is independent of grid resolution, which

can be successively refined or coarsened to study the spatial convergence and computational scalability of the DG2, FV1 and ACC solvers on multi-core CPU and GPU architectures.

As specified by Néelz and Pender (2013), the test is initialised on a rectangular 1000 m × 2000 m flat, dry floodplain with a standard grid spacing of $\Delta x = 5$m. A semi-circular flood wave emanates from a narrow, 20 m breach at the centre of the western boundary as given by the inflow discharge hydrograph shown in Fig. 5b. The test is ended after 5 hours. Manning's

coefficient $n_M$ is fixed at $0.05 \, \mathrm{sm}^{-1/3}$ leading to Froude numbers below 0.25, making the test suitable for all solvers including LISFLOOD-ACC. For each solver, water depth and velocity hydrographs are measured at four standard gauge point locations marked in Fig. 5a, and the water depth cross-section is measured after 1 hour along the centre of the domain at y = 1000 m.

### 3.1.1  Water depth and velocity hydrographs

Predicted hydrographs are obtained for the ACC, FV1-CPU, FV1-GPU, DG2-CPU and DG2-GPU solvers (Fig. 6). FV1-CPU

and FV1-GPU solutions are identical and are named collectively as FV1 (similarly, DG2-CPU and DG2-GPU are named collectively as DG2). While no exact solution is available, DG2, FV1 and ACC predictions of water depth and velocity agree closely with existing industrial model results (Fig. 4.10 and 4.11 in Néelz and Pender (2013)). ACC and DG2 water depth predictions are almost identical at all gauge points (Fig. 6a–d). FV1 predictions are nearly identical, except that the wave front is slightly smoother and arrives several minutes earlier than ACC or DG2, as seen at point 5 (Fig. 6c) and point 6 (Fig. 6d).

Differences in velocity predictions are more pronounced (Fig. 6e–h). The biggest differences are seen at point 1 (Fig. 6e), located only 50 m from the breach, since the flow at this point is dominated by strong inflow discharge with negligible retardation by frictional forces. At point 1, ACC and DG2 velocity predictions agree closely with the majority of industrial models (Fig. 4.11 in Néelz and Pender (2013)). LISFLOOD-FV1 predicts faster velocities up to $0.5 \, \mathrm{ms}^{-1}$, which is close to the prediction of TUFLOW FV1 (Huxley et al., 2017, Table 11). Further away from the breach at point 3 (Fig. 6f), point 5 (Fig. 6g) and point

---

[1]Rain-on-grid features have since been added to the optimised ACC solver, and will be available in a future LISFLOOD-FP release.

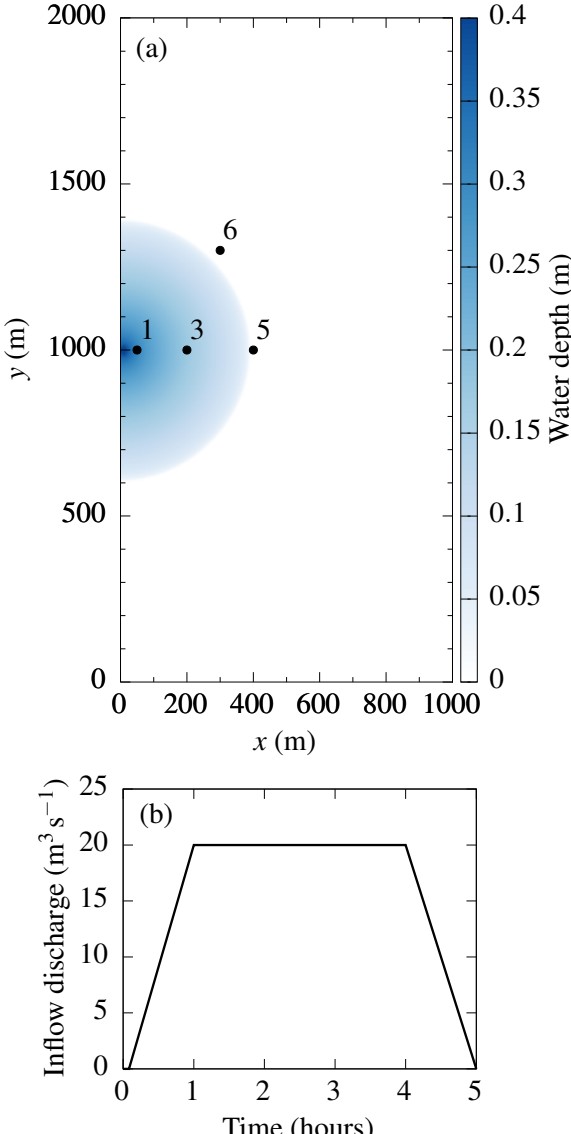

**Figure 5.** (a) Semi-circular flood wave after one hour, with the locations of gauge points 1, 3, 5 and 6 marked. (b) Trapezoidal inflow discharge hydrograph with a peak flow of $20\,\mathrm{m}^3\mathrm{s}^{-1}$.

6 (Fig. 6h), velocity predictions agree more closely, except at the time of wave arrival. At this time, DG2 predicts the sharpest velocity variations while ACC velocity predictions are slightly smoother. FV1 predicts even smoother velocity variations with slightly lower peak velocities.

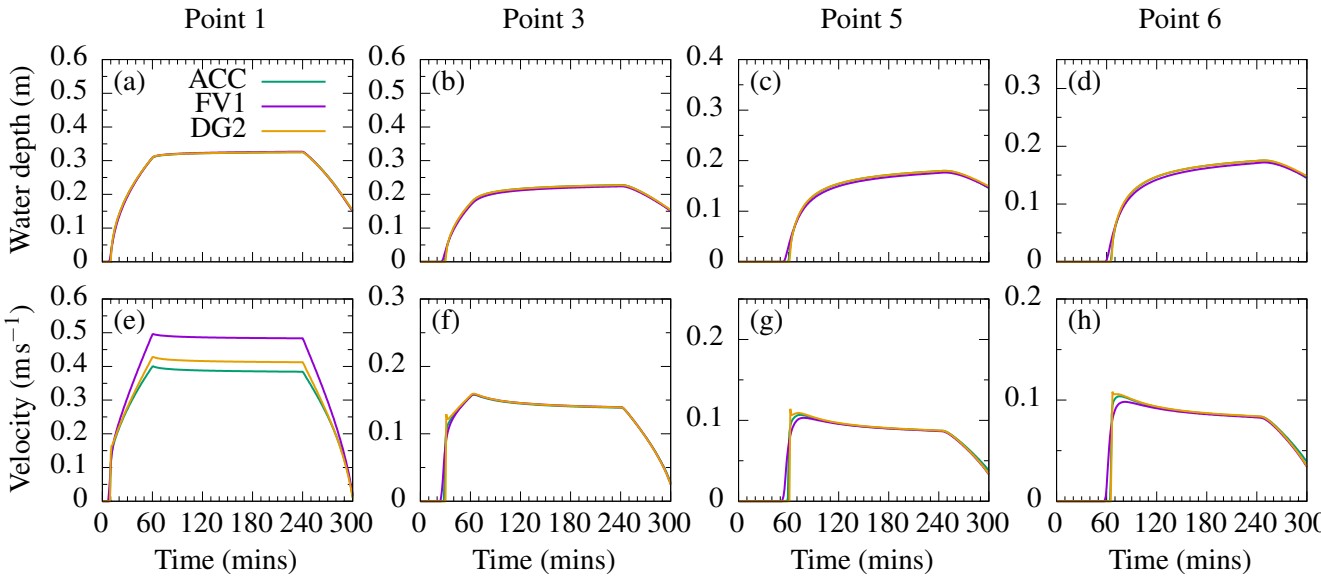

**Figure 6.** ACC, FV1 and DG2 predictions of water depth and velocity hydrographs at gauge points 1, 3, 5 and 6, using the standard grid spacing of $\Delta x = 5$ m.

### 3.1.2 Spatial grid convergence

Spatial grid convergence is studied by modelling at grid resolutions of $\Delta x = 5$ m, 1 m, and 0.5 m. Since the floodplain is flat,
no topographic resampling is required. On each grid, the water depth cross-section is measured along the centre of the domain
(Fig 7). DG2, FV1 and ACC cross-sectional profiles at the standard grid spacing of $\Delta x = 5$ m agree well with industrial
model results (Fig 4.13 in Néelz and Pender (2013)). Differences are most apparent in the vicinity of the wave-front, near
$x = 400$ m. At the standard resolution of $\Delta x = 5$ m, FV1 predicts a wave-front about 50 m ahead of ACC or DG2, and the FV1
solution is much smoother. A TUFLOW modelling study reported similar findings, with TUFLOW FV1 predicting a smoother
wave-front about 50 m ahead of other TUFLOW solvers (Huxley et al., 2017, Table 12). At a five-times finer resolution of
$\Delta x = 1$ m, all solvers predict a steeper wave-front, although the FV1 wave-front prediction at $\Delta x = 1$ m is still relatively
smooth, being closer to the ACC prediction at $\Delta x = 5$ m. A ten-times finer resolution of $\Delta x = 0.5$ m is required for FV1 to
predict a steep wave-front in agreement with DG2 at $\Delta x = 5$ m, while ACC only requires a resolution of $\Delta x = 2$ m to obtain
similar agreement.

These differences can be attributed to the order-of-accuracy of the solvers: DG2 is formally second-order accurate and
exhibits the least sensitivity to grid resolution; FV1 is formally first-order accurate and exhibits the greatest sensitivity, with
numerical diffusion errors leading to a spuriously smooth wave. Despite its simplified formulation, ACC predictions are close
to DG2 because ACC is second-order-accurate in space (Sect. 2.3).

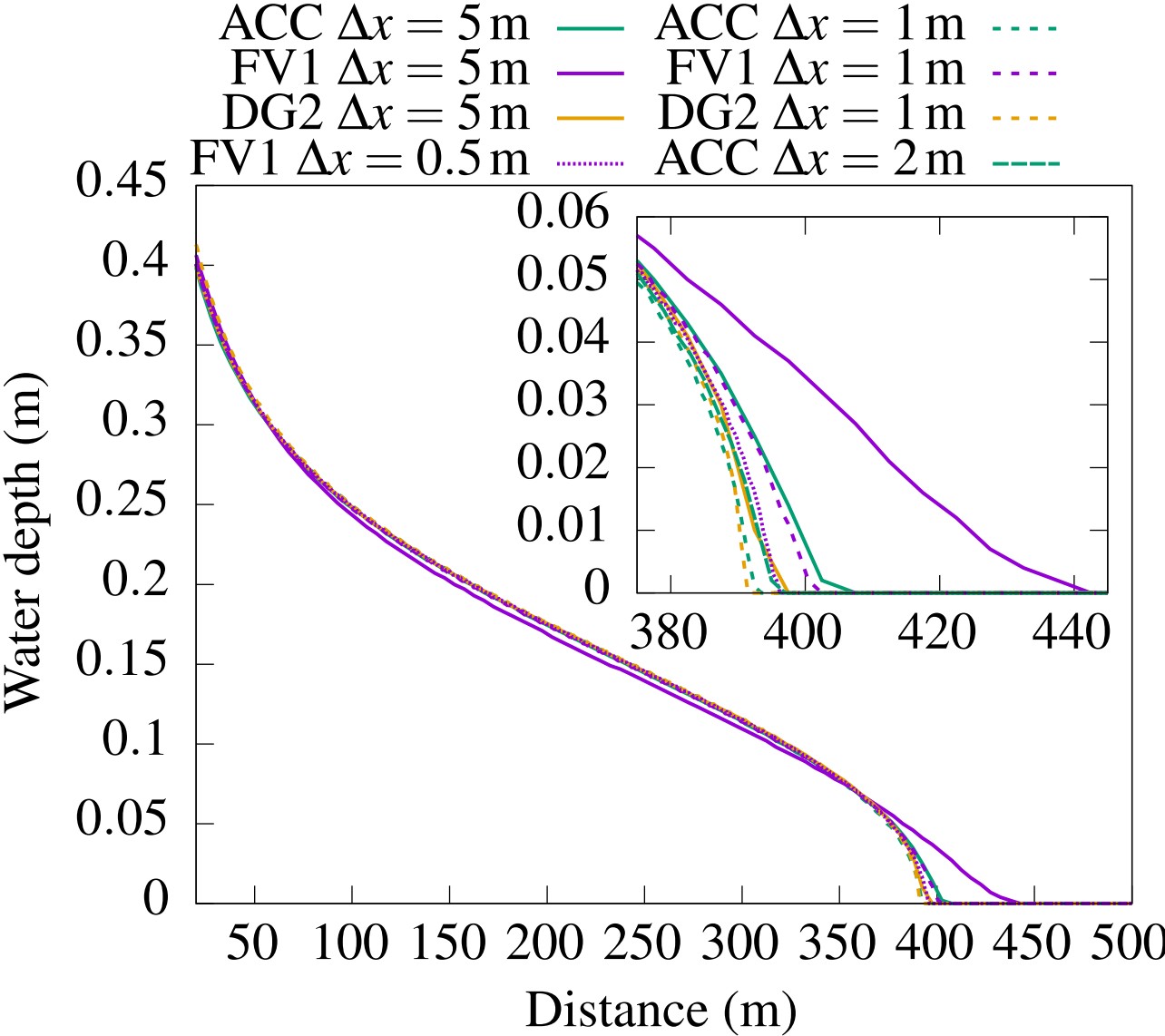

**Figure 7.** ACC, FV1 and DG2 water depth cross-sections after 1 hour. The inset panel shows the wave-front profile across a zoomed-in portion of the cross-section.

### 3.1.3 Solver runtimes for a varying number of elements

To assess the relative runtime cost of the solvers, the test is run for a range of grid resolutions from $\Delta x = 10$ m (yielding $2 \times 10^4$ elements) to $\Delta x = 0.5$ m ($8 \times 10^6$ elements). Each of the ACC, FV1-CPU and DG2-CPU solvers are run using 16 CPU cores, and FV1-GPU and DG2-GPU are run on a single GPU. To ensure reliable measurements, each solver is run twice on each grid, and the fastest runtime is recorded. Runs that have not completed within 24 hours are aborted and excluded from the results. Solver runtimes are shown in Fig. 8a on a log-log scale.

On the coarsest grid with $2 \times 10^4$ elements, FV1-CPU and FV1-GPU both take 5 seconds to complete—just 2 seconds more than ACC. As the grid is refined and the number of elements increases, FV1-CPU remains slightly slower than ACC, while FV1-GPU becomes faster than ACC when $\Delta x < 5$ m and the number of elements exceeds $10^5$. The runtime cost relative to ACC is shown in Fig. 8b: FV1-CPU is about 1.5–2.5× slower than ACC, gradually becoming less efficient as the number of elements increases. In contrast, FV1-GPU becomes about 2× faster than ACC (relative runtime $\approx 0.5$) once the number of

elements exceeds $10^6$ ($\Delta x \sim 1$ m), when the high degree of GPU parallelisation is exploited most effectively.

   Similar trends are found with DG2-CPU and DG2-GPU: on the coarsest grid DG2-CPU is about twice as fast as DG2-GPU, but DG2-GPU becomes increasingly efficient as the number of elements increases, being twice as fast as DG2-CPU at $\Delta x = 2$ m with $5 \times 10^5$ elements (Fig. 8c). At $\Delta x = 1$ m with $2 \times 10^6$ total elements, DG2-GPU completes in about 3.5 hours while the DG2-CPU run is aborted, having failed to complete within 24 hours (Fig. 8a).

As seen earlier in the inset panel of Fig. 7, similar wave-fronts were predicted by DG2 at $\Delta x = 5$ m, ACC at $\Delta x = 2$ m, and FV1 at $\Delta x = 0.5$ m. At these resolutions, DG2-CPU, DG2-GPU and ACC achieved a similar solution quality for a similar runtime cost, with all solvers completing in about 4 minutes (Fig. 8a). Meanwhile, the DG2 solvers on a ten-times coarser grid were 140× faster than FV1-CPU (10 hours 42 minutes) and 28× faster than FV1-GPU (1 hour 47 minutes).

### 3.1.4 Multi-core CPU scalability

To assess the computational scalability of the multi-core CPU solvers, the test is run using 1–16 CPU cores, while FV1-GPU and DG2-GPU are run on a single GPU device. A grid spacing of $\Delta x = 2$ m ($5 \times 10^5$ elements) is chosen so that the grid has sufficient elements for effective GPU parallelisation (informed by the GPU runtimes in Fig. 8b–c), but has few enough elements so that all model runs complete within the 24 hour cutoff. Measured runtimes for ACC, FV1-CPU and DG2-CPU are shown in Fig. 9 on a log-log scale, with all solver runtimes decreasing as the number of CPU cores increases. To facilitate

a like-for-like comparison with FV1 and DG2, ACC solver runtimes were obtained for the ACC implementation of Neal et al. (2012a). Theoretical 'perfect scaling' lines are marked by thin dotted lines for each solver: perfect scaling means that doubling the number of CPU cores would halve the runtime. ACC solver scalability falls somewhat below perfect scaling, with a 16-fold increase in CPU cores only yielding a 7-fold decrease in runtime (Fig. 9). In contrast, the DG2-CPU and FV1-CPU solvers achieve close-to-perfect scaling up to 4 CPU cores, with synchronisation overheads causing only a small decrease in

scalability thereafter. It is expected that additional performance can be gained by using the alternative, CPU-optimised ACC implementation (Neal et al., 2018), and these CPU-specific optimisations are also under consideration for future enhancement

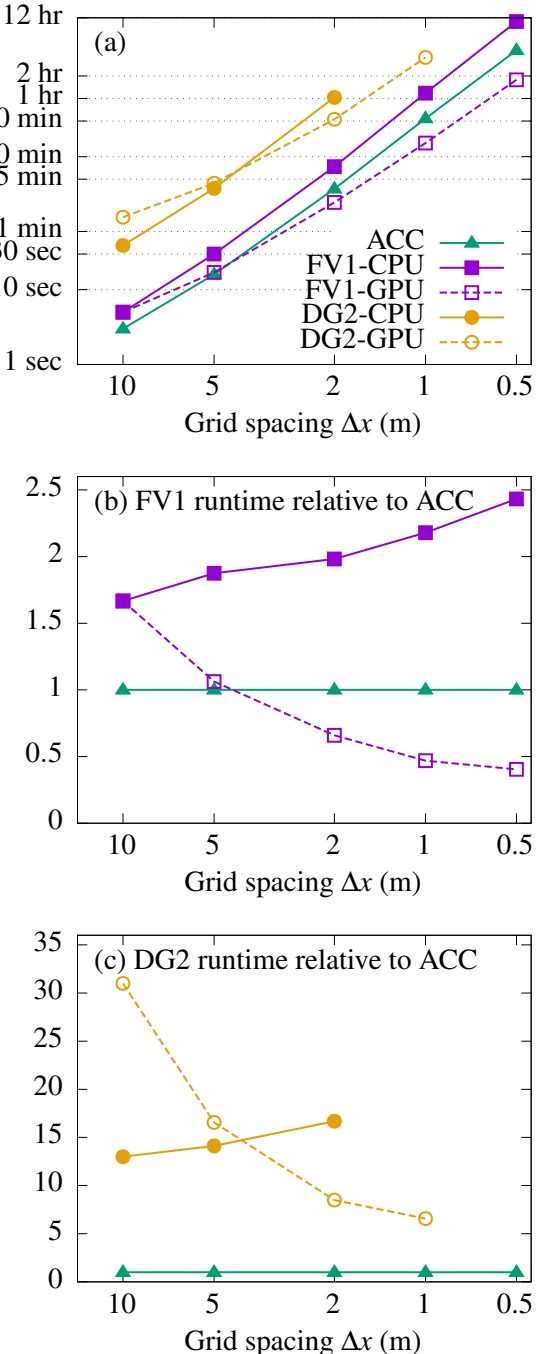

**Figure 8.** (a) Solver runtimes at $\Delta x = 10$ m ($2 \times 10^4$ total elements), $\Delta x = 5$ m ($8 \times 10^4$ elements), $\Delta x = 2$ m ($5 \times 10^5$ elements), $\Delta x = 1$ m ($2 \times 10^6$ elements), and $\Delta x = 0.5$ m ($8 \times 10^6$ elements). The ACC, FV1-CPU and DG2-CPU solvers are run on a 16-core CPU while the FV1-GPU and DG2-GPU solvers are run on a single GPU. Runtimes are presented relative to ACC for (b) FV1 and (c) DG2: values greater than one represent a slow-down relative to ACC; values less than one represent a speed-up relative to ACC. ACC solver runtimes were obtained for the ACC implementation of Neal et al. (2012a). **19**

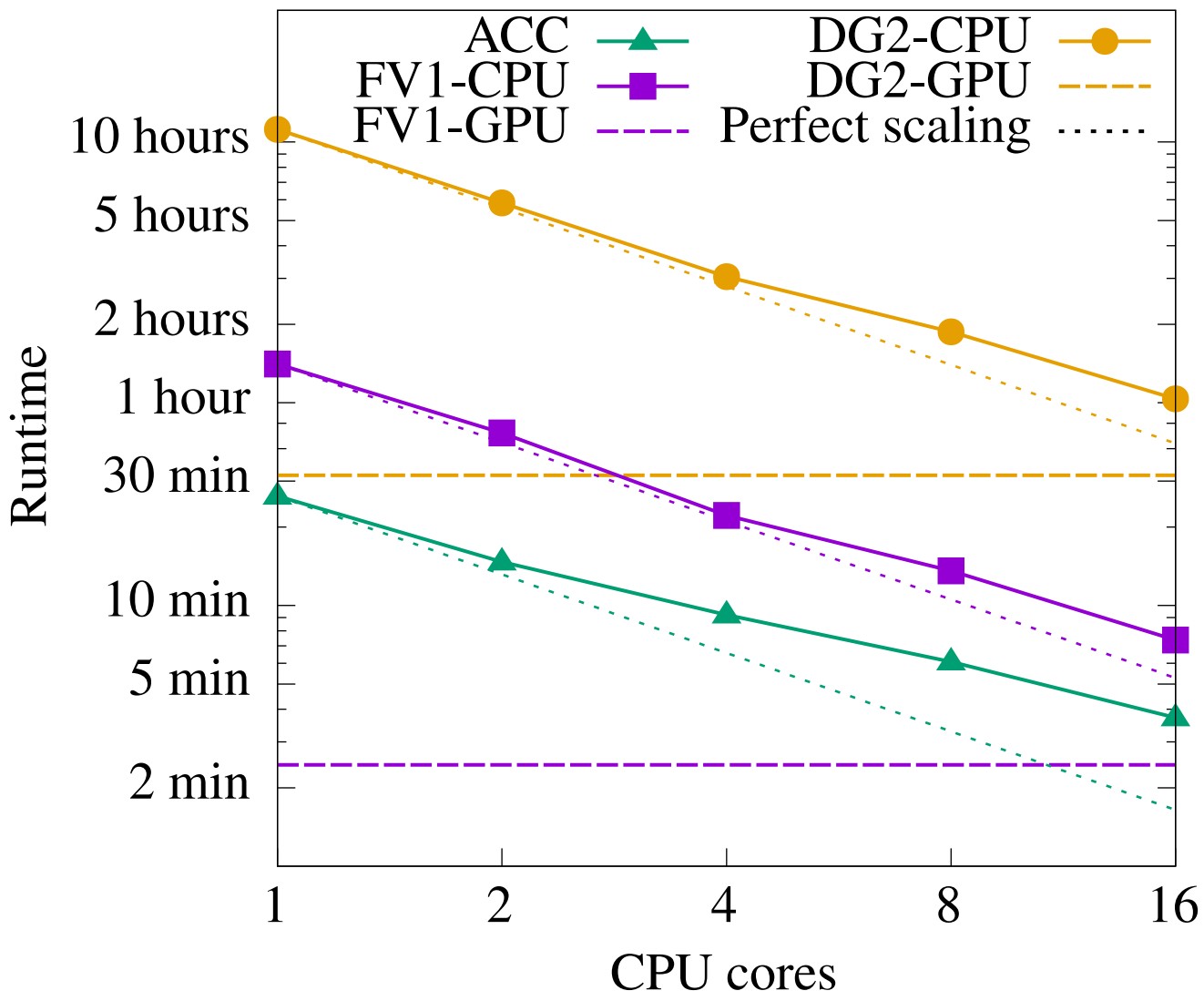

**Figure 9.** ACC, FV1-CPU and DG2-CPU solver runtimes for test 4 on a grid with 500,000 elements (at $\Delta x = 2$ m) using 1–16 CPU cores. The theoretical perfect scaling of each solver—doubling the number of CPU cores halves the runtime—is marked by thin dotted lines. FV1-GPU and DG2-GPU runtimes are marked by dashed horizontal lines (the number of GPU cores is not configurable). ACC solver runtimes were obtained for the ACC implementation of Neal et al. (2012a).

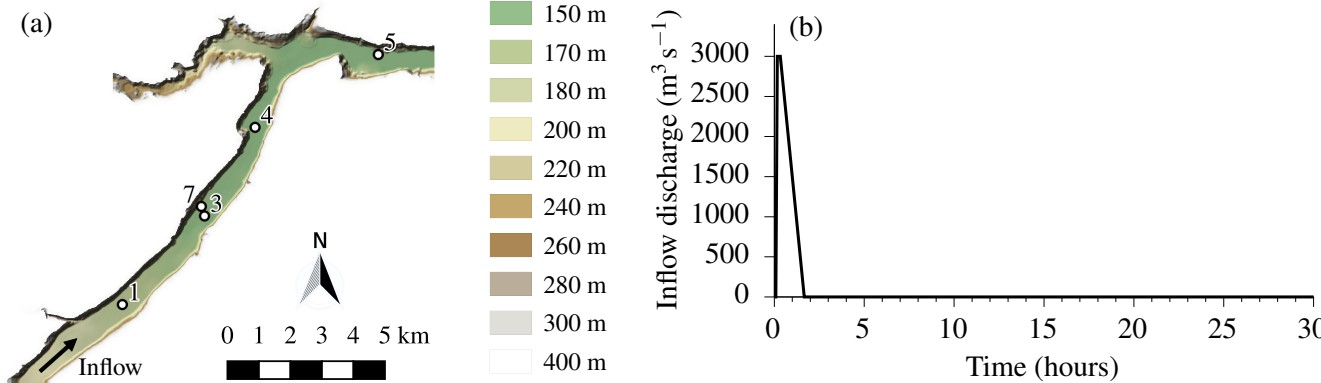

**Figure 10.** Configuration of the rapidly-propagating flow test: (a) Terrain elevation map, with the positions of gauge points 1, 3, 4, 5 and 7 marked; (b) Prescribed inflow discharge hydrograph with a skewed trapezoidal profile over the first 100 minutes of the 30 hour simulation.

of the DG2 and FV1 solvers. For intercomparison with the CPU solvers, FV1-GPU and DG2-GPU runtimes are also marked by dashed horizontal lines (since the number of GPU cores is not configurable). Both GPU solvers are substantially faster than their counterparts on a 16-core CPU.

Overall, the FV1, ACC and DG2 solvers converged on similar water depth solutions with successive grid refinement. Owing to its first-order accuracy, FV1 requires a very fine resolution grid to match the solution quality of DG2 or ACC, though FV1-GPU enables runtimes up to 6× faster than the 16-core FV1-CPU solver. Thanks to its second-order accuracy, DG2 water depth predictions are spatially converged at coarser resolutions (Fig. 7). Hence, DG2 is able to replicate the modelling quality of FV1 at a much coarser resolution, and the multi-core DG2-CPU solver is a competitive choice for grids with fewer than 385   100,000 elements.

### 3.2   Rapidly-propagating wave along a valley

This test, known as Test 5 in Néelz and Pender (2013), is employed to assess the capabilities of the DG2, FV1 and ACC solvers for modelling rapidly-propagating flow over realistic terrain. As specified by Néelz and Pender (2013), the narrow valley (Fig. 10a) is initially dry, and Manning's coefficient $n_M$ is fixed at $0.04\ \mathrm{sm}^{-1/3}$. A synthetic dam break event near the 390   southern boundary is modelled by prescribing a short inflow discharge hydrograph along a 260 m-long line near the southern edge of the domain, with a peak flow of $3000\ \mathrm{m}^3\mathrm{s}^{-1}$ (Fig. 10b). The test is ended after 30 hours once the water has ponded near the closed boundary at the eastern edge of the domain.

     LISFLOOD-FP is run using the ACC, FV1 (CPU and GPU), and DG2 (CPU and GPU) solvers at the standard DEM resolution of $\Delta x = 50$ m used in most existing studies (Cohen et al., 2016; Huxley et al., 2017; Neal et al., 2018), and at the 395   finest available DEM resolution of $\Delta x = 10$ m. Water level and velocity hydrographs are measured at the five standard gauge point locations marked in Fig. 10a.

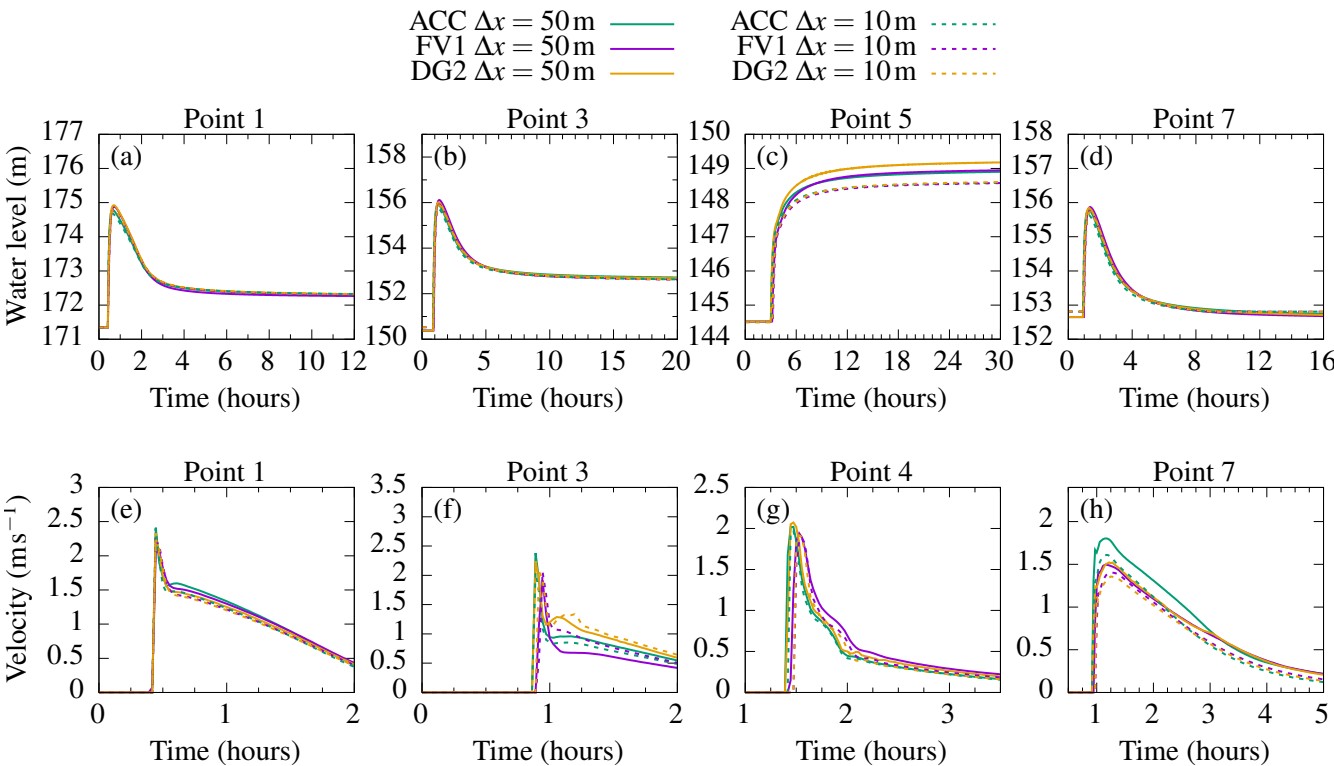

**Figure 11.** ACC, FV1 and DG2 predictions of water level and velocity hydrographs at gauge points 1, 3, 4 (velocity only), 5 (water level only) and 7, using the standard resolution of $\Delta x = 50$ m, and finest resolution of $\Delta x = 10$ m.

### 3.2.1 Water level and velocity hydrographs

Predicted water level and velocity hydrographs are shown in Fig. 11. The water level hydrographs show that water ponds in small topographic depressions at point 1 (Fig. 11a), point 3 (Fig. 11b) and point 5 (Fig. 11c). Point 7 is positioned near the steep valley slope, and is only inundated between $t = 1$ hour and $t = 8$ hours (Fig. 11d). At both resolutions, water levels predicted by all solvers agree closely with existing industrial model results at points 1, 3 and 7 (Fig. 4.16 in Néelz and Pender (2013)). Small water level differences accumulate as water flows downstream and at point 5, positioned farthest downstream of the dam break, differences of about 0.5 m are found depending on the choice of resolution and solver (Fig. 11c). Similar water level differences have been found amongst the suite of TUFLOW solvers (Huxley et al., 2017) and amongst other industrial models (Néelz and Pender, 2013).

Bigger differences are found in velocity predictions, particularly at locations farther downstream at point 3 (Fig. 11f), point 4 (Fig. 11g) and point 7 (Fig. 11h). At point 3, DG2 predicts small, transient velocity variations at $\Delta x = 50$ m starting at $t = 1$ hour; these variations are not captured by the FV1 or ACC solvers, but have been captured by a FV2-MUSCL solver at the finest resolution of $\Delta x = 10$ m, as reported by Ayog et al. (2021). At point 7, ACC overpredicts peak velocities by about

$0.5 \text{ ms}^{-1}$ compared to FV1 and DG2 (Fig. 11h), and compared to other industrial models (Fig. 4.17 in Néelz and Pender (2013)). Otherwise, ACC, FV1 and DG2 velocity predictions are within the range of existing industrial model predictions.

### 3.2.2 Flood inundation and Froude number maps

While hydrograph predictions are often studied for this test case (Néelz and Pender, 2013; Cohen et al., 2016; Huxley et al., 2017; Neal et al., 2018), flood inundation maps taken at time instants provide a more detailed picture of flood wave propagation. Accordingly, two sets of flood inundation maps are obtained: one set at $t = 15$ minutes during the short period of peak inflow, and another set at $t = 3$ hours once the flood water has almost filled the valley. Flood maps are obtained at the finest resolution of $\Delta x = 10$ m, and with DG2 at $\Delta x = 50$ m.

After 15 minutes, water has travelled about 1.5 km north-east along the valley away from the inflow region near the southern edge of the domain, with ACC, FV1 and DG2 water depth predictions shown in Fig. 12a–d. Behind the wave-front, an abrupt change in water depth is predicted by FV1 (Fig. 12b) and DG2 (Fig. 12c, d), but this discontinuity induces spurious, small-scale oscillations in the ACC solver that propagate downstream (Fig. 12a). This numerical instability is understood by studying the Froude number, as shown in Fig. 12e–h. The rapidly-propagating flow becomes supercritical across the region of shallower water, with a maximum Froude number of around 1.5. The local inertia equations are not physically valid for modelling abrupt changes in water depth or supercritical flows (de Almeida and Bates, 2013), leading to the observed numerical instability in the ACC solver.

After 3 hours, the flood water has filled most of the valley and the wave-front has almost reached point 5. As shown in Fig. 12i–l, ACC, FV1 and DG2 water depth predictions are in close agreement. The flow is now predominantly subcritical (Fig. 12m–p), though a small region of supercritical flow is found upstream of point 3 with a maximum Froude number of about 1.2 and a corresponding jump in water depth at the same location. Nevertheless, numerical instabilities in the ACC prediction at $t = 15$ mins are no longer evident at $t = 3$ hours (Fig. 12m), and ACC predictions remain stable at all gauge points for the duration of the simulation (Fig. 11). As seen in the fourth column of Fig. 12, DG2 flood maps at $\Delta x = 50$ m are in close agreement with the ACC, FV1 and DG2 flood maps at $\Delta x = 10$ m.

### 3.2.3 Runtime cost

Simulation runtimes are summarised in Table 1, with FV1-CPU and DG2-CPU solvers run on a 16-core CPU, and FV1-GPU and DG2-GPU solvers run on a single GPU. Similar to runtime measurements presented earlier in Sect. 3.1.3, the GPU solvers become more efficient on grids with a larger number of elements: in this test, DG2-GPU is $1.8\times$ faster than DG2-CPU at the standard resolution of $\Delta x = 50$ m, becoming $2.5\times$ faster at the finest resolution of $\Delta x = 10$ m; similarly, FV1-GPU is between $1.2\times$ and $5.1\times$ faster than FV1-CPU.

DG2-CPU and DG2-GPU at $\Delta x = 50$ m outperform ACC, FV1-CPU and FV1-GPU at $\Delta x = 10$ m, while still achieving similarly accurate flood map predictions at a $5\times$ coarser resolution (Fig. 12). DG2-CPU at $\Delta x = 50$ m is $2\times$ faster than ACC at $\Delta x = 10$ m, while DG2-GPU is twice as fast again. DG2-GPU flood maps at an improved resolution of $\Delta x = 20$ m are obtained at a runtime cost of 38 mins, which is still competitive with ACC at $\Delta x = 10$ m (with a runtime cost of 30 mins).

**Figure 12.** Water depth and Froude number maps of the rapidly-propagating wave along a valley: (a–h) after 15 minutes across a zoomed-in portion of the domain near the dam break; (i–p) after 3 hours across the entire domain. The entire simulation is ended after 30 hours once the water has ponded near at the eastern edge of the domain. Water depth colour scales vary between $t = 15$ minutes and $t = 3$ hours, but Froude number colour scales remain fixed.

**Table 1.** Solver runtimes at grid spacings of $\Delta x = 50$ m, $\Delta x = 20$ m, and $\Delta x = 10$ m. ACC, FV1-CPU and DG2-CPU solvers are run on a 16-core CPU; FV1-GPU and DG2-GPU solvers are run on a single GPU. ACC solver runtimes were obtained for the ACC implementation of Neal et al. (2012a).

| | $\Delta x = 50$ m 57 000 elements | $\Delta x = 20$ m 850 000 elements | $\Delta x = 10$ m 1.7 million elements |
|---|---|---|---|
| ACC | 20 s | 466 s (8 mins) | 1779 s (30 mins) |
| FV1-CPU | 22 s | 739 s (12 mins) | 2188 s (36 mins) |
| FV1-GPU | 19 s | 145 s (2 mins) | 965 s (16 mins) |
| DG2-CPU | 788 s (13 mins) | 4133 s (69 mins) | 33009 s (9 hours) |
| DG2-GPU | 448 s (7 mins) | 2304 s (38 mins) | 13606 s (4 hours) |

In summary, all solvers predicted similar water depth and velocity hydrographs, though ACC experienced a short period of numerical instability in a localised region where the Froude number exceeded the limit of the local inertia equations. The shock-capturing FV1 and DG2 shallow water solvers yield robust predictions throughout the entire simulation. with FV1-GPU being consistently faster than ACC on a 16-core CPU. As found earlier in Sect. 3.1.3, DG2 at a 2–5× coarser resolution is a competitive alternative to ACC and FV1, with the GPU implementation being preferable when running DG2 on a grid with more than 100,000 elements.

### 3.3 Catchment-scale rain-on-grid simulation

In December 2015, Storm Desmond caused extensive fluvial flooding across the Eden catchment in North West England (Szönyi et al., 2016). This storm event has previously been simulated using a first-order finite volume hydrodynamic model (Xia et al., 2019), with overland flow and fluvial flooding driven entirely by spatially- and temporally-varying rainfall data over the 2500 $km^2$ catchment. As such, this simulation is ideally-suited to assess the new rain-on-grid capabilities in LISFLOOD-FP 8.0, and represents one of the first DG2 hydrodynamic modelling studies of rainfall-induced overland flow across a large catchment. At this large scale, grid coarsening is often desirable to ensure model runtimes remain feasible (Falter et al., 2013), but coarsening the DEM can affect the quality of water depth predictions (Savage et al., 2016). Therefore, the three LISFLOOD-FP solvers were run at a range of grid resolutions, and their predictions were analysed with respect to observed river levels and flood extent survey data.

The Eden catchment and its four major rivers are shown in Fig. 13a. The DEM is available at a finest resolution of $\Delta x = 5$ m covering the entire catchment. The largest city in the Eden catchment is Carlisle, situated at the confluence of the River Irthing, Petteril, Caldew and Eden (Fig. 13b). In the Carlisle area, the 5 m DEM incorporates channel cross-section and flood defence data (Xia et al., 2019), and manual hydro-conditioning to remove bridge decks that would otherwise block river flows.

As specified by Xia et al. (2019), the simulation comprises a spin-up phase and subsequent analysis phase. The spin-up phase starts at 00:00 3 December 2015 from an initially dry domain. Water is introduced into the domain via the rainfall source term (Eqn. 14), using Met Office rainfall radar data at a 1 km resolution updated every 5 minutes (Met Office, 2013). The

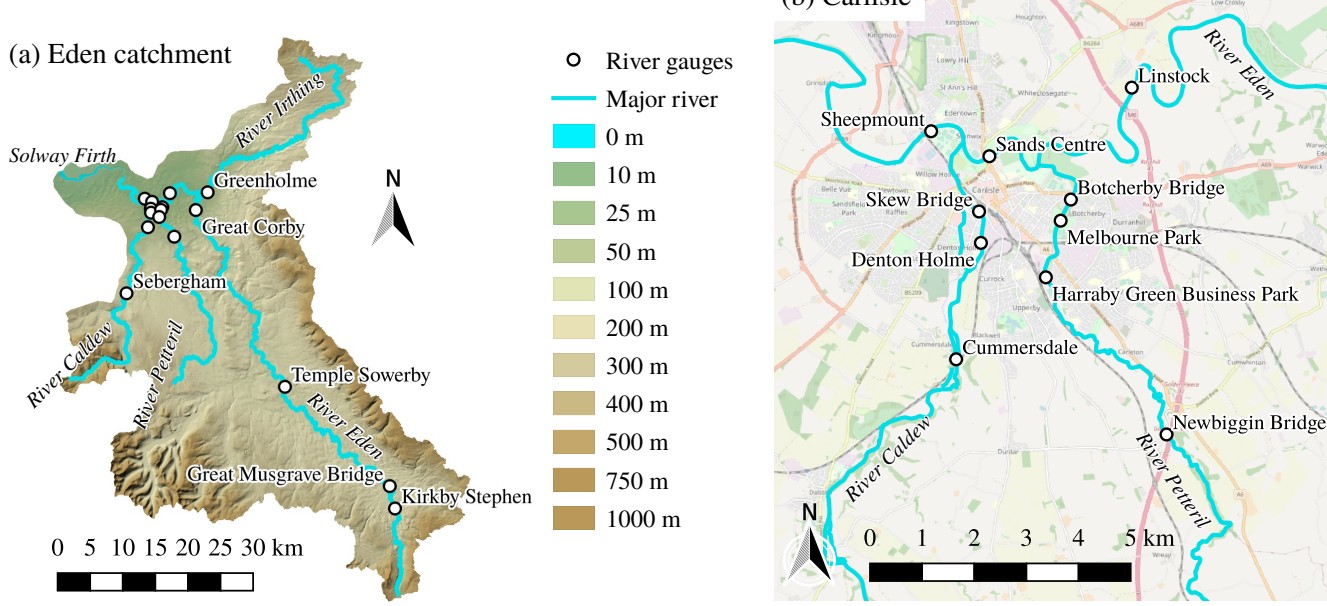

**Figure 13.** Elevation map of (a) the Eden catchment, covering an area of 2500 km², and (b) a zoomed-in portion over Carlisle at the confluence of the River Irthing, Petteril, Caldew and Eden. Names and locations of the sixteen gauging stations are marked. Contains Ordnance Survey data © Crown copyright and database right 2020. © OpenStreetMap contributors 2020. Distributed under a Creative Commons BY-SA License.

spin-up phase ends and the analysis phase begins at 12:00 4 December 2015, once the memory of the dry initial condition has disappeared, and water depths and discharges in all river channels have reached a physically realistic initial state (Xia et al., 2019). The simulation ends at 12:00 8 December 2015 after a total of 5.5 simulated days. Manning's coefficient $n_M$ is 0.035 sm$^{-1/3}$ for river channels and 0.075 sm$^{-1/3}$ elsewhere.

The following modelling assumptions are made as specified by Xia et al. (2019). Zero infiltration is assumed due to fully-saturated antecedent soil moisture. An open boundary condition is imposed along the irregular-shaped catchment perimeter by adjusting the terrain elevation of elements lying outside the catchment such that their elevation is below mean sea level, thereby allowing water to drain out of the River Eden into the Solway Firth. At each time-step, water flowing out of the Solway Firth is removed by zeroing the water depth in elements lying outside the catchment. While rainfall data errors can influence model

outputs, Ming et al. (2020) found that a prescribed 10% rainfall error lead to only 5% relative mean error in predicted water depth hydrographs. As such, modelling uncertainties due to rainfall errors are not quantified in these deterministic model runs.

     Model input data is prepared by upscaling the finest 5 m DEM to resolutions of $\Delta x = 40$ m, 20 m and 10 m. In previous studies, a grid spacing of $\Delta x = 10$ m was sufficient to simulate observed flood extent and river levels (Xia et al., 2019; Ming et al., 2020), so LISFLOOD-FP runs are not performed on the finest 5 m DEM. Given the large number of elements (25 million

elements at $\Delta x = 10$ m) and informed by the computational scalability results in Sect. 3.1.3 and 3.2.3, DG2 and FV1 runs are

**Table 2.** Manually-adjusted gauging station positions given in British National Grid (EPSG:27700) coordinates. Terrain elevation error is measured as the local elevation difference between the 40 m DEM and 10 m DEM. Channel widths are also estimated at each gauging station using the finest resolution DEM.

| Gauging station | Easting (m) | Northing (m) | Terrain elevation error (m) | Estimated channel width (m) |
|---|---|---|---|---|
| Sheepmount | 338940 | 557120 | 1.91 | 67 |
| Sands Centre | 340203 | 556650 | 1.63 | 56 |
| Linstock | 342868 | 557869 | 2.00 | 71 |
| Great Corby | 346770 | 555440 | 1.56 | 54 |
| Skew Bridge | 339949 | 555519 | 5.01 | 15 |
| Denton Holme | 339971 | 554898 | 1.76 | 15 |
| Botcherby Bridge | 341656 | 555778 | 1.33 | 9 |
| Melbourne Park | 341462 | 555397 | 1.14 | 11 |
| Cummersdale | 339492 | 552682 | 2.30 | 14 |
| Newbiggin Bridge | 343473 | 551360 | 2.00 | 8 |
| Greenholme | 348575 | 558071 | 2.23 | 21 |
| Harraby Green Business Park | 341160 | 554379 | 1.12 | 9 |
| Sebergham | 336193 | 542590 | 6.06 | 13 |
| Temple Sowerby | 360444 | 528379 | 1.70 | 33 |
| Great Musgrave Bridge | 376445 | 513112 | 4.60 | 31 |
| Kirkby Stephen | 377283 | 509772 | 1.37 | 8 |

only performed on a GPU, while ACC is run on a 16-core CPU. Due to its relatively high runtime cost, DG2-GPU is only run at $\Delta x = 40$ m.

For each model run, hydrographs of free-surface elevation above mean sea level are measured in river channels at sixteen gauging stations as marked in Fig. 13. Approximate gauging station coordinates are provided by Environment Agency (2020), but these are often positioned near the river bank and not in the channel itself. Hence, gauging station coordinates must be adjusted to ensure model results are measured in the channel. Here, a simple approach is adopted to manually reposition each gauging station based on the finest resolution DEM, with amended positions given in Table 2. It is also important to measure hydrographs of free-surface elevation, since variation in free-surface elevation is minimal across the river channel. DG2, FV1 and ACC solver predictions are compared in the following three subsections: first, predicted free-surface elevation hydrographs are compared against gauging station measurements; second, predicted maximum flood extents are compared against a post-event flood extent survey (McCall, 2016); finally, predicted flood inundation maps are intercompared.

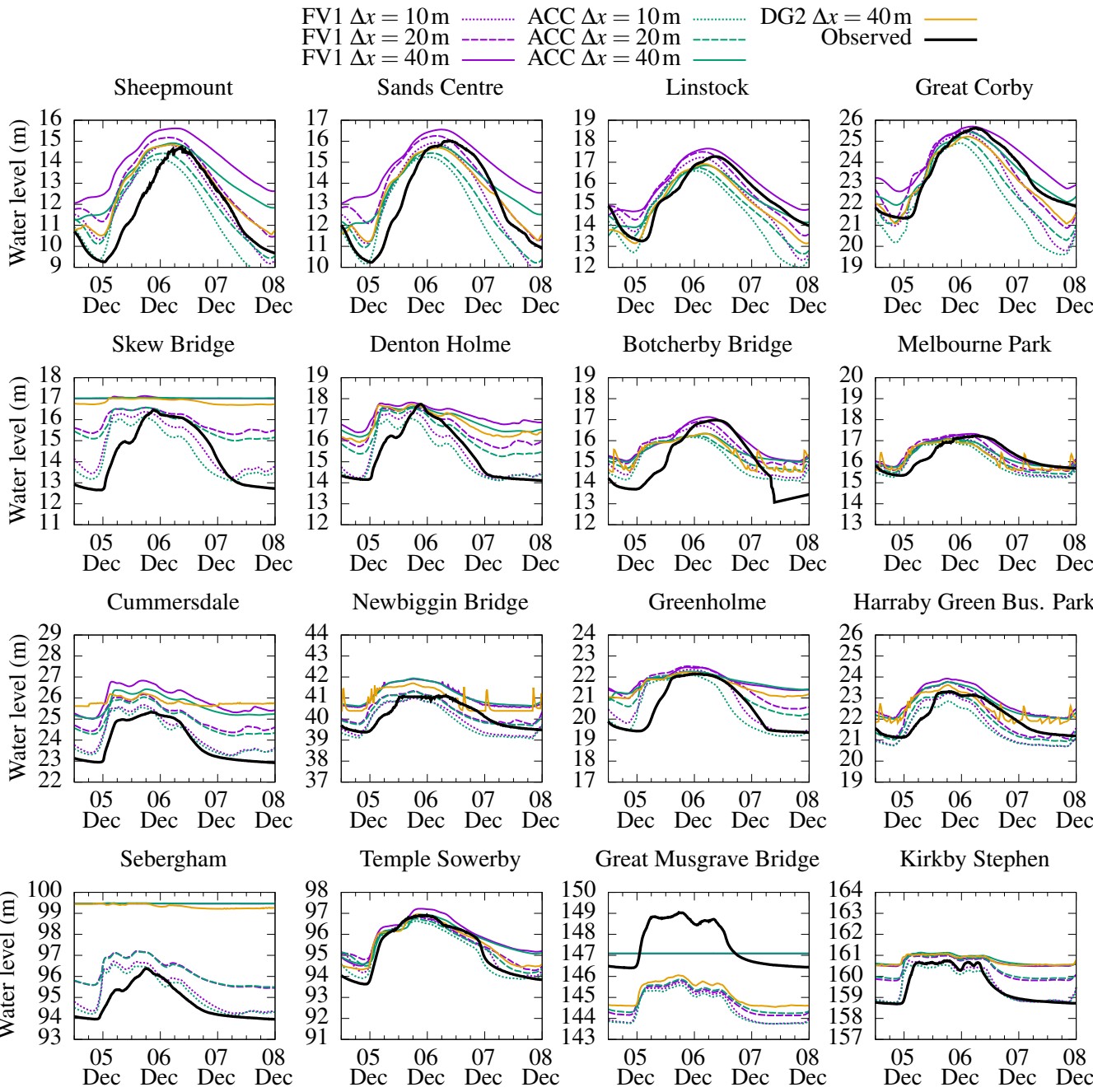

**Figure 14.** Hydrographs of free-surface elevation at sixteen river gauging stations, as shown in Fig. 13. Observed hydrographs are marked by thick black lines; model predictions are marked by coloured lines.

### 3.3.1 River channel free-surface elevation hydrographs

Free-surface elevation hydrographs at the sixteen river gauging stations are shown in Fig. 14. Observed free-surface elevation hydrographs are calculated from Environment Agency measurements of water depth and river bed elevation above mean sea level (Environment Agency, 2020). While water depths can be measured to an accuracy of $\sim 0.01$ m (Bates et al., 2014), discrepancies between in-situ, point-wise river bed elevation measurements and the remotely-sensed, two-dimensional DEM can result in systematically biased free-surface elevations, as reported by Xia et al. (2019).

As seen in the observed hydrographs, river levels begin to rise across the catchment around 00:00 5 December. A flashy response is seen in the headwaters of the River Eden, at Temple Sowerby, Great Musgrave Bridge, and Kirkby Stephen, with water levels rising rapidly by 2–3 m, and returning almost as rapidly to base flow conditions around 00:00 7 December. Similar responses are found at the other gauging stations located further downstream, where river levels vary more gradually. The largest river level changes are found in the Carlisle area, particularly at Sheepmount and Sands Centre, which are located farthest downstream.

Timings of the rising and falling limbs are well-predicted by all three solvers for the majority of hydrographs. At coarser grid resolutions, river levels are overpredicted, and the difference between base flow and peak flow levels is underpredicted[2]. These findings are consistent with those of Xia et al. (2019). Hydrograph inaccuracies are primarily due to DEM coarsening, which artificially smooths river channel geometries, reducing the elevation difference between river bed and river bank. Consequently, the terrain elevation at gauging points on the 40 m DEM are between 1.12 m and 6.06 m higher than the same points on the 10 m DEM, depending on the local river channel geometry. These terrain elevation errors are shown in Table 2, which are calculated as the difference in local element-average topography elevations between the 40 m DEM and 10 m DEM.

The impact of DEM coarsening is most evident at Sebergham gauging station where the largest terrain elevation error of 6.06 m is found. At $\Delta x = 40$ m, the DEM diverts the flow away from the true location of the river, and the FV1 and ACC Sebergham hydrographs remain flat at 99.4 m. At $\Delta x = 20$ m, the terrain is only 1.4 m higher than at $\Delta x = 10$ m and the FV1 and ACC hydrographs are closer to observations, though the difference between base flow and peak flow levels is still underpredicted. At $\Delta x = 10$ m, predicted hydrographs accurately capture observed base flow and peak flow levels. The same behaviour is evident at Skew Bridge (with a terrain elevation error of 5.01 m) and, to a lesser extent, at other locations including Cummersdale (2.30 m) and Greenholme (2.23 m). In general, the greater the terrain elevation error at a given point, the greater the discrepancy between observed hydrographs and model predictions.

Next, the predictive capability of DG2 on the coarsest grid is benchmarked against hydrograph observations, and against FV1 and ACC predictions on the 4× finer grid. To measure the average discrepancy between predictions and observations, the

---

[2]Except for anomalous predictions found at Great Musgrave Bridge, where the observed hydrograph shape is generally well-captured but free-surface elevations are consistently underpredicted. This anomaly is due to localised terrain elevation differences between the finest resolution DEM and Environment Agency river bed elevation measurements, as documented by Xia et al. (2019).

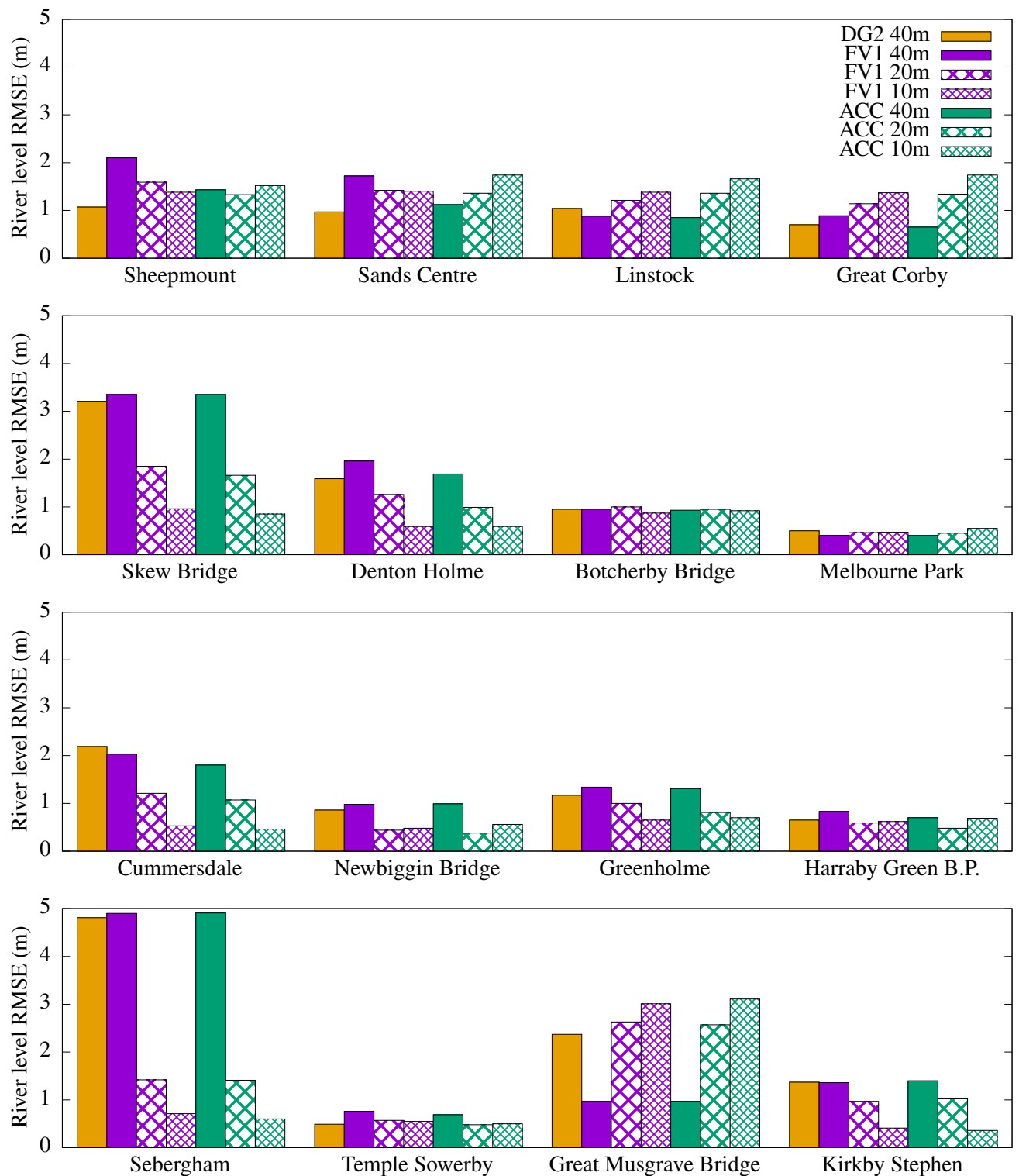

**Figure 15.** Root mean square errors in predicted free-surface elevation hydrographs, with errors measured against observation data.

RMSE is calculated as

$$\text{RMSE} = \sqrt{\frac{\sum_{t_{\text{start}}}^{t_{\text{end}}} \left[ \left( z_{i,j,0} + h_{i,j,0}^n \right) - \eta_{\text{obs}}^n \right]^2}{N}} \tag{20}$$

where $t_{\text{start}} = 12\!:\!00$ 4 December, $t_{\text{end}} = 00\!:\!00$ 8 December, $\eta_{\text{obs}}^n$ is the free-surface elevation calculated from observation data, and $N$ is the total number of observations. At most gauging stations, predictions converge towards observations, with

RMSEs becoming smaller as the grid is refined. But at some gauging stations, including Linstock and Great Corby, the falling limb is underpredicted on finer grids, so RMSEs increase as predictions diverge from observations. Similar behaviour was also found at some gauging stations in the original study of Xia et al. (2019).

At most gauging stations, DG2 alleviates the free-surface elevation overprediction found in FV1 and ACC hydrographs at the same resolution, leading to better agreement between DG2 predictions and observations at Sheepmount, Sands Centre, Skew

Bridge, Denton Holme, Greenholme and Sebergham, as indicated by the RMSEs in Fig. 15. The reduced overprediction is attributable to DG2's locally-planar representation of terrain within each computational element, which enables DG2 to better capture terrain gradients between the river bed and river bank on a coarse grid.

DG2 predictions are also closer to FV1 and ACC hydrographs on $2\times$ and $4\times$ finer grids, depending on the river width at each gauging station, which ranges between 8 m and 71 m (Table 2). The widest locations are at Sheepmount, Sands Centre,

Linstock, Great Corby, Temple Sowerby and Great Musgrave Bridge; locations with moderate river widths of 13 m–21 m are found at Denton Holme, Greenholme, Skew Bridge, Sebergham; at most other locations rivers are narrower. At the widest locations, DG2 predictions are close to FV1 and ACC hydrographs on the $4\times$ finer grid; at locations with moderate river widths, DG2 predictions are closer to FV1 and ACC hydrographs on the $2\times$ finer grid. At other locations, DG2 predictions are closer to FV1 and ACC hydrographs at the coarsest grid resolution. Overall, when river channel geometries are larger than

$\Delta x/2$ then the predictive capability of DG2 is substantially enhanced thanks to its second-order accurate, piecewise-planar representation of terrain and flow variables.

Where river channel widths are close to or smaller than the grid spacing $\Delta x$, hydrograph predictions are especially sensitive to the channel geometry as resolved on the computational grid. At such locations, hydrograph predictions can be improved by running the model with an ensemble of possible sampling positions within a 100 m radius of each gauging station, then

choosing the best fit between predictions and observations. However, this approach relies on the availability of observation data and, due to modelling sensitivities at the scale of the grid, optimal positions can vary depending on the choice of solver and grid resolution. Spatially-adaptive solvers (Kesserwani and Sharifian, 2020; Özgen-Xian et al., 2020) and non-uniform meshing techniques (Kolega and Syme, 2019) offer another alternative to improve flow predictions by selectively capturing fine-scale channel geometries, and such methods are under development for inclusion in a future LISFLOOD-FP release. Subgrid channel

modelling can also improve hydrograph and flood inundation predictions, and LISFLOOD-FP already provides a sub-grid channel model (Neal et al., 2012a) that could be integrated with the DG2 and FV1 solvers in a future release.

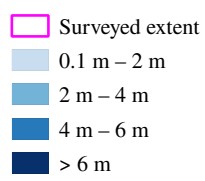

Surveyed extent
0.1 m – 2 m
2 m – 4 m
4 m – 6 m
> 6 m

**Figure 16.** Maximum flood extent predictions compared against the post-event surveyed extent outlined in pink. DG2 predictions at $\Delta x = 20$ m and $\Delta x = 10$ m are downscaled from the DG2 piecewise-planar prediction at $\Delta x = 40$ m. Arrows mark the most notable differences in maximum water depth, as discussed in Sect. 3.3.2.

**Table 3.** Hit rate (H), false alarm ratio (F) and critical success index (C) for the DG2 and FV1 predictions of maximum flood extent calculated against the reference solution of the ACC solver at $\Delta x = 10$ m.

| | $\Delta x = 40$ m | | | $\Delta x = 20$ m | | | $\Delta x = 10$ m | | |
|---|---|---|---|---|---|---|---|---|---|
| | H | F | C | H | F | C | H | F | C |
| DG2 | 0.83 | 0.32 | 0.59 | 0.86 | 0.55 | 0.40 | 0.85 | 0.56 | 0.40 |
| FV1 | 0.77 | 0.27 | 0.59 | 0.80 | 0.19 | 0.67 | 0.93 | 0.03 | 0.90 |

### 3.3.2 Maximum flood extent over Carlisle

Maximum flood extents are obtained for ACC and FV1 runs at resolutions of $\Delta x = 40$ m, 20 m, and 10 m; due to its relatively high runtime cost, DG2-GPU is only run at $\Delta x = 40$ m only, with flood maps at 20 m and 10 m being inferred by downscaling the 40 m solution. The downscaling procedure adopted here exploits the full, DG2 piecewise-planar solution by constructing the piecewise-planar maximum water depth on the $\Delta x = 40$ m grid, then sampling at the element centres on the higher-resolution grid.

As shown in Fig. 16, the post-event survey outlined in pink marks the maximum extent of flooding across Carlisle. The surveyed flood extent is well-predicted by all solvers. Predicted flood extents are largely insensitive to grid resolution, except for the region around Denton Holme gauging station on the River Caldew, which is protected by flood defence walls. Xia et al. (2019) added these flood defence walls by hand in the original 5 m DEM, but the coarsened DEMs were upscaled with no further hand-editing. As such, the steep, narrow walls become smeared out at coarse resolutions, with all solvers overpredicting flood extents at $\Delta x = 40$ m in the Denton Holme region. The representation of these flood defences could be improved by adopting the recently-developed LISFLOOD-FP levee module[3] (Wing et al., 2019; Shustikova et al., 2020), or by implementing a spatially-adaptive multi-resolution method that selectively refines the grid resolution around river channels and other fine-scale features (Kesserwani and Sharifian, 2020).

Further qualitative differences are apparent in predicted water depths in regions south of Linstock and north of Botcherby Bridge, as indicated by arrows in Fig. 16. At $\Delta x = 40$ m, DG2 and ACC yield almost identical predictions with regions of 0.1–2 m water depth south of Linstock and depths of 2–4 m north of Botcherby Bridge. In contrast, FV1 predicts wider areas of water depths of 2–4 m south of Linstock and depths of 4–6 m north of Botcherby Bridge. These regions of deep water become smaller as the grid is refined, but FV1 flood inundation predictions remain wider and deeper than ACC even at $\Delta x = 10$ m.

The DG2 and FV1 predictions of maximum flood extent can be quantified against the ACC prediction at $\Delta x = 10$ m, which is treated as the reference solution. The hit rate measures flood extent underprediction as the proportion of wet elements in the reference solution that were also predicted as wet. The false alarm ratio measures flood extent overprediction as the proportion of predicted wet elements that were dry in the reference solution. The critical success index measures both over- and underprediction. All three metrics range between 0 and 1, and further details are provided by Wing et al. (2017).

---

[3]Not yet available with the FV1 or DG2 solvers.

**Table 4.** Root Mean Square Error (RMSE) in water depth (m) calculated at 12:00 5 December over the entire Eden catchment. The FV1 prediction at the finest resolution of $\Delta x = 10$ m is taken as the reference solution. RMSEs are not calculated for DG2 at $\Delta x = 20$ m or 10 m because these results are downsampled from the DG2 $\Delta x = 40$ m result.

|  | DG2 | FV1 | ACC |
|---|---|---|---|
| $\Delta x = 40$ m | 0.241 | 0.267 | 0.240 |
| $\Delta x = 20$ m | — | 0.112 | 0.100 |
| $\Delta x = 10$ m | — | — | 0.037 |

The hit rate (H), false alarm ratio (F) and critical success index (C) are given in Table 3. At $\Delta x = 40$ m, the critical success index is 0.59 for both DG2 and FV1, but DG2 has a higher hit rate and false alarm ratio, suggesting that DG2 predicts a wider flood extent than ACC or FV1. At $\Delta x = 20$ m and $\Delta x = 10$ m, the false alarm ratio and critical success index for DG2 deteriorate, but a hit rate of 0.83–0.86 is maintained, which is acceptable given that high-resolution predictions are downscaled from the DG2 piecewise-planar solution at $\Delta x = 40$ m. FV1 predictions at $\Delta x = 20$ m and $\Delta x = 10$ m are obtained directly without downscaling, and FV1 predictions converge towards ACC predictions with successive grid refinement. This convergence is evidenced in all three metrics, with FV1 at $\Delta x = 10$ m achieving a high hit rate (0.93), low false alarm ratio (0.03), and high critical success index (0.90).

### 3.3.3 Flood inundation maps at 12:00 5 December

While some differences between solver predictions were evident in maximum flood depths, these differences become clearer in flood inundation maps taken at a single time instant. Accordingly, flood inundation maps shown in Fig. 17 are taken at 12:00 5 December over Carlisle city centre, during the rising limb of the Sheepmount and Sands Centre hydrographs, where river level rises were largest (Fig. 14). At the coarsest resolution of $\Delta x = 40$ m, DG2 and ACC predictions are almost identical (Fig. 17a and 17g). Both solvers accurately capture the flood extent and water depths predicted by FV1 and ACC at a $4\times$ finer resolution of $\Delta x = 10$ m (Fig. 17f and 17i). In contrast, FV1 predicts greater water depths and a slightly wider flood extent, particularly at coarser resolutions of $\Delta x = 40$ m (Fig. 17d) and $\Delta x = 20$ m (Fig. 17e). But once the grid is refined to a resolution of $\Delta x = 10$ m, FV1 and ACC solutions are almost converged (Fig. 17f and 17i). DG2 predictions at $\Delta x = 20$ m (Fig. 17b) and 10 m (Fig. 17c) are downscaled from the DG2 prediction at $\Delta x = 40$ m. The downscaled DG2 predictions are not expected to resolve all fine-scale features visible in the FV1 and ACC predictions. Nevertheless, compared to the DG2 $\Delta x = 40$ m flood map, the downscaled DG2 flood maps better represent the deeper waters in the River Eden (flowing east-to-west), and in the River Caldew (flowing south-to-north).

To quantify the spatial convergence of the three solvers, water depth RMSEs are calculated at 12:00 5 December over the entire catchment (Table 4). Since water depth observations are unavailable, the FV1 prediction at $\Delta x = 10$ m is taken as the reference solution. At $\Delta x = 40$ m, DG2 and ACC RMSEs are almost identical, while the FV1 error is about 10% larger. At

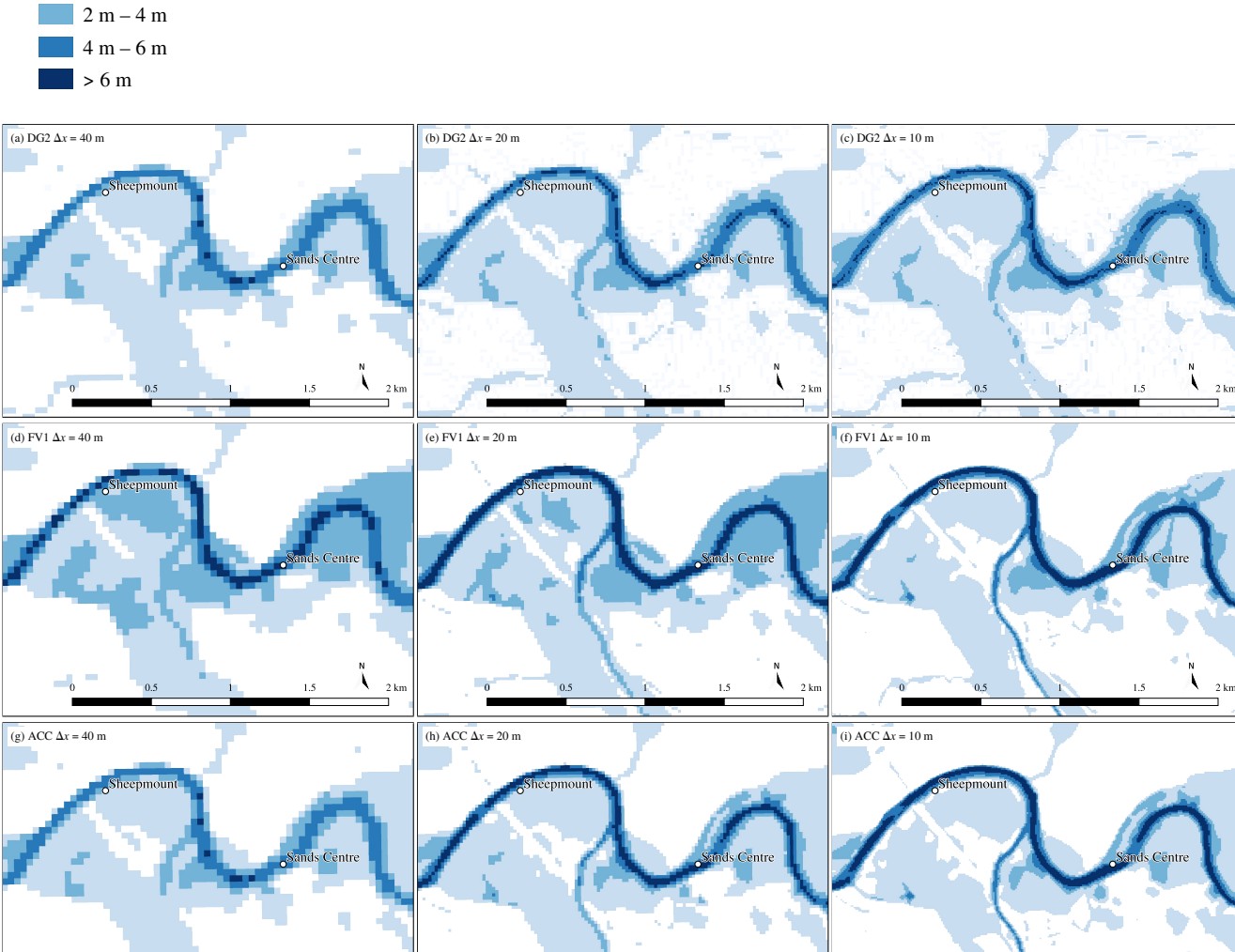

**Figure 17.** Predicted flood inundation maps over Carlisle city centre at 12:00 5 December. DG2 predictions at $\Delta x = 20$ m and $\Delta x = 10$ m are downscaled from the DG2 piecewise-planar prediction at $\Delta x = 40$ m.

**Table 5.** Solver runtimes for DG2-GPU, FV1-GPU and ACC solvers. The ACC solver is run on 16 CPU cores. Due to its relatively high runtime cost, DG2-GPU is only run at $\Delta x = 40$ m.

|  | DG2-GPU | FV1-GPU | ACC |
|---|---|---|---|
| $\Delta x = 40$ m | 154 hours | 1.2 hours | 2.4 hours |
| $\Delta x = 20$ m | — | 8.1 hours | 17.5 hours |
| $\Delta x = 10$ m | — | 67 hours | 131 hours |

$\Delta x = 20$ m, FV1 errors are again about 10% larger than ACC, with the ACC solver converging more rapidly towards the FV1 reference solution than FV1 itself, despite ACC's simplified numerical formulation (Sect. 2.3).

In this catchment-scale simulation and earlier in the simulation of a slowly-propagating wave over a flat floodplain (Sect. 3.1.2), FV1 was seen to converge more slowly and predict a flood extent wider than DG2 or ACC. Once again, these differences can be attributed to the order-of-accuracy of the solvers: FV1 is formally first-order accurate and exhibits the greatest sensitivity to grid resolution, while DG2 and ACC are both second-order-accurate in space.

### 3.3.4 Runtime cost

Solver runtimes for the entire 5.5-day simulation are shown in Table 5. On the same grid, FV1-GPU is about $2\times$ faster than ACC on a 16-core CPU, which is consistent with earlier findings in Sect. 3.1 and 3.2. FV1-GPU and ACC runtimes scale as expected: halving the grid spacing quadruples the total number of elements and halves the time-step due to the CFL constraint. Hence, halving the grid spacing multiplies the runtime by a factor of about eight. At all grid spacings between 40 m and 10 m, FV1-GPU and ACC simulations run faster than real-time and complete in less than 5.5 days, indicating that these solvers are suitable for real-time flood forecasting applications.

The DG2-GPU solver runtime is substantially slower than other solvers on the same, coarse grid. Unlike the tests presented earlier in Sect. 3.1 and 3.2, this test involves widespread overland flow driven by continual rainfall. Overland flow is characterised by thin layers of water only centimetres deep, which continually flow downhill, driven by gravity and balanced by frictional forces. These frictional forces become strongly nonlinear for such thin water layers, and the DG2 friction scheme imposes an additional restriction on the time-step size to maintain stability in the discharge slope coefficients. This challenge has recently been addressed in finite volume hydrodynamic modelling using an improved friction scheme that calculates the physically-correct equilibrium state between gravitational and frictional forces (Xia and Liang, 2018). Extending this friction scheme into a discontinuous Galerkin formulation is expected to alleviate the time-step restriction and reduce DG2 solver runtimes for overland flow simulations. For simulations without widespread overland flow, including those presented earlier in Sect. 3.1 and 3.2, the current DG2 formulation imposes no additional time-step restriction and DG2 solver runtimes are substantially faster.

## 4 Summary and conclusions

This paper presented new second-order discontinuous Galerkin (LISFLOOD-DG2) and first-order finite volume (LISFLOOD-FV1) solvers that are parallelised for multi-core CPU and Nvidia GPU architectures. The new solvers are compatible with existing LISFLOOD-FP test cases and available in LISFLOOD-FP 8.0, alongside the existing local inertia solver, LISFLOOD-ACC. LISFLOOD-FP 8.0 also supports spatially- and temporally-varying rainfall data to enable real-world rain-on-grid simulations.

The predictive capabilities and computational scalability of the new solvers was studied across two Environment Agency (EA) benchmark tests, and for a real-world fluvial flood simulation driven by rainfall across a $2500\ \mathrm{km}^2$ catchment. The second-order spatial convergence of LISFLOOD-DG2 on coarse grids was demonstrated by its ability to sharply resolve moving wet-dry fronts in EA benchmark tests, and in the catchment-scale flood simulation, DG2 alleviated the impact of DEM coarsening on river level hydrograph predictions due to its second-order, piecewise-planar representation of river channel geometries.

By analysing the LISFLOOD-ACC local inertia solver, its hybrid finite-difference/finite-volume scheme was found to be spatially second-order-accurate thanks to its grid staggering of water depth and discharge variables. As a result, ACC predictions in all tests were close to those of DG2, despite ACC's simplified governing equations and simplified numerical scheme. The ACC solver also exhibited less numerical diffusion at wet-dry fronts, and predicted more accurate hydrographs and flood inundation maps than FV1 on coarse grids. Meanwhile, the FV1 and DG2 solvers provided the most robust predictions of a rapidly-propagating wave in an EA benchmark test involving supercritical flow and abrupt water depth changes.

The multi-core FV1-CPU and DG2-CPU demonstrated near-optimal computational scalability up to 16 CPU cores. Multi-core CPU runtimes were most efficient on grids with fewer than 0.1 million elements, while FV1-GPU and DG2-GPU solvers were most efficient on grids with more than 1 million elements, where the high degree of GPU parallelisation was best exploited. On such grids, GPU solvers were 2.5–4× faster than the corresponding 16-core CPU solvers, and FV1-GPU runtimes were highly competitive with those of ACC. DG2-GPU was also found to be more efficient that FV1-GPU and ACC: DG2-GPU delivered the same level of accuracy on 2–4× coarser grids while remaining faster to run.

For the catchment-scale flood simulation, the DG2-GPU runtime was less competitive due to widespread overland flow, involving frictional forces acting on thin water layers, which imposed an additional time-step restriction in the current DG2 formulation. It is expected that this restriction could be lifted by formulating an improved DG2 friction scheme based on the finite volume friction scheme of Xia and Liang (2018). Overland flow does not feature in the EA benchmark tests, where DG2-GPU runtimes remain competitive, being only 5–8× slower than ACC on the same grid. However, FV1 and DG2 are the first solvers in LISFLOOD-FP to gain a dynamic rain-on-grid capability, with this capability being added to the optimised ACC solver in a future release. To further improve efficiency and accuracy at coarse resolutions over large catchments, one future direction would be to port the sub-grid channel model—currently integrated with the CPU-optimised ACC solver—to GPU architectures. Another useful direction would be to enable a multi-resolution solver based on Kesserwani and Sharifian (2020), and introduce a hybrid DG2/FV1 solver that downgraded the DG2 formulation to FV1 in regions of very thin water

layer, or in regions of finest grid resolution, to further reduce the computational cost. Both directions are being investigated for inclusion in future LISFLOOD-FP releases.

Overall, the LISFLOOD-DG2, FV1 and ACC solvers all demonstrated reliable predictions in good agreement with existing industrial model results and real-world observation data. Despite its simplified numerical formulation, ACC predictions were close to those of DG2 since both solvers are spatially second-order-accurate. DG2 achieved the best spatial convergence, and its piecewise-planar representation of river channels wider than $\Delta x/2$ facilitated improved river level hydrograph and flood inundation predictions that were typically close to those of FV1 and ACC on 2–4$\times$ finer grids. Hence, for simulations where

high-resolution DEM data is unavailable or large-scale high-resolution modelling is infeasible, LISFLOOD-DG2-GPU is a promising choice for flood inundation modelling.

*Code and data availability.* LISFLOOD-FP 8.0 source code (LISFLOOD-FP developers, 2020) and simulation results (Shaw et al., 2021) are available on Zenodo. Instructions for running the simulations are provided in Appendix A. Due to access restrictions, readers are invited to contact the Environment Agency for access to the DEM used in Sect. 3.2, and to refer to Xia et al. (2019) for access to the Eden catchment

model data used in Sect. 3.3.

## Appendix A: Running the LISFLOOD-FP simulations

To run a simulation, specify the LISFLOOD-FP parameter file, `ea4.par`, `ea5.par`, or `eden.par` along with the appropriate solver parameters. For example, to run test 4 at $\Delta x = 50$ m with the FV1-GPU solver:

```
lisflood -DEMfile ea4-50m.dem \
-dirroot ea4-50m-fv1-gpu \
         -fv1 -cuda ea4.par
```

Model outputs are written in ESRI ASCII format to the specified `dirroot` directory: `.wd` is a water depth field, and `.Vx` and `.Vy` denote $u$ and $v$ components of velocity. Water depth and velocity hydrographs are written to `.stage` and `.velocity` files respectively.

Model output ESRI ASCII files can be postprocessed using the Python 3 scripts in the `postprocess` directory in the LISFLOOD-FP 8.0 software package (LISFLOOD-FP developers, 2020):

**downsample.py and upsample.py** Downsample or upsample a given ESRI ASCII file by power of two.

**speed.py** Calculate the magnitude of velocity from $u$ and $v$ components.

**froude.py** Calculate the Froude number from given water depth and speed files.

**sampleline.py** Extract a horizontal or vertical cross-section at a given $i$ or $j$ index.

**mask.py** Mask a model output by imposing 'NoData' values from the DEM onto the model output file.

**diff.py** Calculate the difference between two model outputs.

**stats.py** Calculate global statistics including min and max values, and root mean square error.

To convert a raw DEM (`.dem.raw` file) to a DG2 DEM (comprising `.dem`, `.dem1x` and `.dem1y` files), run the `generateDG2DEM` application provided with the LISFLOOD-FP 8.0 software package. For further details on configuring and running the model, consult the user manual (LISFLOOD-FP developers, 2020).

## Appendix B: LISFLOOD-ACC order-of-accuracy

The formal order-of-accuracy of LISFLOOD-ACC is determined by a numerical analysis of the discrete local inertia equations (de Almeida et al., 2012). To begin, the local inertia equations are linearised by assuming small perturbations in free-surface elevation $\eta$ about a constant reference depth $H$ [L], leading to the linearised frictionless one-dimensional local inertia equations:

$$\frac{\partial \eta}{\partial t} + \frac{\partial q}{\partial x} = 0, \tag{B1a}$$

$$\frac{\partial q}{\partial t} + gH\frac{\partial \eta}{\partial x} = 0, \tag{B1b}$$

This linear assumption is valid for gradually-varying, quasi-steady flows (de Almeida et al., 2012), and ensures that the remainder of the analysis is tractable. Eqn. (B1) is then discretised using the same staggered-grid finite-difference approximation as Eqn. (19), before performing a Taylor series expansion of the discrete equations to obtain (de Almeida et al., 2012)

$$\frac{\partial \eta}{\partial t} + \frac{\partial q}{\partial x} = -\left(\frac{1}{2}\frac{\partial^2 \eta}{\partial t^2}\Delta t + \frac{1}{6}\frac{\partial^3 \eta}{\partial t^3}\Delta t^2 + \frac{1}{24}\frac{\partial^3 q}{\partial x^3}\Delta x^2\right), \tag{B2a}$$

$$\frac{\partial q}{\partial t} + gH\frac{\partial \eta}{\partial x} = -\left(\frac{1}{2}\frac{\partial^2 q}{\partial t^2}\Delta t + \frac{1}{6}\frac{\partial^3 q}{\partial t^3}\Delta t^2 + gH\frac{1}{24}\frac{\partial^3 \eta}{\partial x^3}\Delta x^2\right), \tag{B2b}$$

where the first- and second-order discretisation error terms appear on the right-hand side, and higher-order terms are neglected. Considering only the leading-order discretisation errors, Eqn. (B2) simplifies to:

$$\frac{\partial \eta}{\partial t} + \frac{\partial q}{\partial x} = \mathcal{O}(\Delta t) + \mathcal{O}(\Delta x^2), \tag{B3a}$$

$$\frac{\partial q}{\partial t} + gH\frac{\partial \eta}{\partial x} = \mathcal{O}(\Delta t) + \mathcal{O}(\Delta x^2), \tag{B3b}$$

where $\mathcal{O}(\cdot)$ denotes the leading-order discretisation errors. Therefore, the LISFLOOD-ACC formulation is formally first-order-accurate in time but second-order-accurate in space.

*Author contributions.* JS coded the numerical solvers in collaboration with JN, and conducted simulations and drafted the initial manuscript with assistance from MKS. GK secured project funding and provided supervision. All authors contributed to conceptualisation, manuscript review and editing.

*Competing interests.* The authors declare that they have no conflict of interest.

*Financial support.* JS, GK and MKS were supported by the UK Engineering and Physical Sciences Research Council (EPSRC) grant 715 EP/R007349/1. PB was supported by a Royal Society Wolfson Research Merit award. This work is part of the SEAMLESS-WAVE project (SoftwarE infrAstructure for Multi-purpose fLood modElling at variouS scaleS based on WAVElets). For information about the project visit https://www.seamlesswave.com.

*Acknowledgements.* The authors are most grateful to Xiaodong Ming (Newcastle University), Qiuhua Liang and Xilin Xia (Loughborough University) for providing model data and valuable expertise for the Storm Desmond Eden catchment simulation. The authors thank Peter 720 Heywood and Robert Chisholm (The University of Sheffield) for sharing their CUDA expertise, and to Janice Ayog (The University of Sheffield) for her expertise with Environment Agency benchmark tests.

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
