# Peer review of "LISFLOOD-FP 8.0: the new discontinuous Galerkin shallow water solver for multi-core CPUs and GPUs"

_Geoscientific Model Development, 2020_

## Referee Comment (RC1) · Anonymous Referee #1 · 17 Nov 2020

Dear authors,

Congratulations to this very clear and concise piece of work. The overall goal, methods, and results of the model development and model application are described with sufficient detail, and the overall structure of the paper supports reading and understanding of the content. Also, the presentation of the results is in most cases clear and informative.

The fact that model code and study results are provided on Zenodo is very much appreciated, great! A short readme or similar would help, however, to guide the user through the folder structure of the repository.

Before publication is possible, there are nevertheless some minor aspects which should be addressed in an updated version of the manuscript.

1. Page 2 / lines 52-43: maybe good to mention the license under which the model is distributed? This gives directly an idea of what is permitted and not. Additionally, I strongly encourage you to add a DOI in the manuscript linking to the Zenodo-version used in this manuscript.
2. Page 11 / line 213: why are these recent optimizations lacking? Please provide a brief explanation.
3. Section 3.1 and 3.2: for both test cases, it is stated that they are widely used to assess model predictions. What I thus would like to see is an extended discussion of how model results here generally compare to other modeling studies. For instance, are the trend presented here and their magnitude comparable? A bit more depth is needed to move the manuscript more towards a scientific publication rather than an extended model documentation logbook.
4. Chapter 3.1.2: Here, the cases are benchmarked for two different resolutions. The initial 5 m resolution and a finer 1 m resolution. Question 1 is, how did you derive the new geographical data, how did you perform the resampling? And question 2, why did you not look into grid coarsening - wouldn't the chance of having abrupt water level changes between cells become larger when applying grids with, admittedly probably very much, coarser spatial resolution? This would be less relevant to assess speed of the solvers, but accuracy.
5. Chapter 3.3: While for the two Environment Agency test cases a motivation was stated, it does not become clear where why exactly this case study was selected. Please elaborate briefly why you decided to use this test case and not another one from the rich literature of LISFLOOD-FP studies, for example.
6. Page 27 / Line 423: this is a very important aspect and should be highlighted more prominently in the manuscript (e.g. abstract and/or summary). One of the key reasons many scholars/practitioners use LISFLOOD-FP is its sub-grid scheme. This also holds for the comment made in point 2.
7. Chapter 3.3.2: For the analyses of flood extent, would it not be useful to include metrics like the hit rate, false alarm ration and critical success index to quantify the actual (dis)agreement between simulations and results?
8. Figure 16 and Figure 17: This figure is hard to read. While adding the OSM background map is appreciated for geographical reference, (as done in a figure above), it's diluting the actual information about the flood maps in the current form. Please consider revising this figure.
9. Summary/Conclusions: this section nicely wraps up the manuscript. However, recommendations for further studies and improvements, and the shortcomings of the current version (both feature-wise and technologically), are missing. Please set your work in context of what was done so far, how your work adds to that and opens up new possibilities, and what challenges are still lying ahead to fast hydrodynamic simulation over coarse and large domains.

---

## Referee Comment (RC2) · Anonymous Referee #2 · 22 Dec 2020

Review of "LISFLOOD-FP 8.0: the new discontinuous Galerkin shallow water solver for multi-core CPUs and GPUs" By Shaw et al.

Major remarks

The inclusion of a second-order DG2 discretisation and implementation in the LISFLOOD-FP 8.0 flood forecasting suite, on a regular mesh, of the shallow water equations with friction and rainfall, for flood forecasting, is presented, including analyses of three test cases and comparison with a local inertia solver ACC and a first-order FV1 discretisation and implementation. Parallelisation and GPU results are intercompared in terms of performance and runtime.

While the paper is very interesting and the work constitutes a useful addition in a flood forecasting suite for public use, which is very important, there seem to be several loose ends and unclear aspects that need to be resolved before publication is warranted. I therefore recommend to return the manuscript for major revisions.

It would be desirable that the following major issues are addressed, either clarified or resolved:

(i) Since ACC, FV1 and DG2 schemes are tested and compared more clarity on their characteristics is desired in order to understand the differences in the results presented. Why is DG2 much slower than FV1 in the third test? The time stepping schemes of FV1 and DG2 are unclear. Figuring out the split scheme for DG2 given rainfall and damping is too hard or impossible also, since the references are unclear; it is easier to simply state the full scheme; there is sufficient space in (9) to do so and remove any ambiguity. The split scheme for Sf is not implicit, once one looks into the references. Why is the handling of friction (Sf) in DG2 leading to troubles for thin, fast overland flow while it is not troubling for FV1 and ACC? Perhaps use the same time stepping scheme in DG2 as for FV1 or make aspects of Sf semi-implicit. Please clarify and make improvements.

(ii) In test 3, but also in earlier tests, DG2 has topography at dx=40 (or another dx for tests 1 and 2) and seemingly the higher degrees of freedom used for the variables is not used for the topography, i.e. the dx=10 topography information can be used to make finer projections onto the topography given that DG2 is second order and the topography should not be planar. Hence, the (at times and in certain simulations) observed simulation underperformance of DG2 seems to be/is more severe than needed.

(iii) DG2 is used on a regular mesh; DG is most optimal for non-uniform situations but this option or restriction is not discussed; also not discussed are hybrid 1D and 2D schemes or hybrid FV1 and DG2 options, the latter which should be easy to accommodate. At least a discussion is warranted.
(iv) Some of the speed and/or accuracy comparisons seem incomplete or unfair: e.g., in Fig. 6, not the same resolution should (only) be compared but (also) the speed for mixed resolutions with (roughly) the same accuracy.

(v) The is a large accumulation of minor remarks, see below, which should be refuted an addressed.

The above major remarks are also reflected, sometimes on several occasions, in several of the minor/detailed remarks given below.

Minor/detailed remarks

Line 23: simplify (plural)

Line 42: But there are plenty of FV2 solvers, which may be faster; has a comparison with those been made or at least discussed? Please discuss.

Line 56: What about mixed 1D and 2D solvers, see the literature?

Line 58: This paper [use of "remainder" is a bit much on page 2].

Line 72: What is new or simplified relative to K2018, and why?
 Please specify

Eqn (2): define x, y, t.

Line 80: define L and T.

Line 83: I don't like calling $C_f$ a coefficient since it includes a dynamic variable, i.e. h; rephrase; it is a friction function. I would take out $h^{1/3}$ (of $C_f$) and even write it is $h^\alpha$ since alpha is a parameterization anyway.

Line 90: Why is this a dot product? No dot; it is a matrix-vector product.

Line 119: there should be a flag doing that (application of a slope limiter) automatically for smooth solutions, not a hand-switch. See Krivodonova or extensions for improvements. In any real flooding case, this should be done via an automatic switch. Please clarify and add a sensible automatised switch. That the test cases do not involve

shock wave propagation, is that not an omission? Tests 2 and 3 must have seen some shock-wave propagation prior to settling into a quasi-steady state, given the F>1, F<1 transition observed and discussed. Please clarify.

Line 121: In KS2020, I do not see thus split implicit scheme explained? Rather KS2020 refers to KL2010, where matters are also not explained, so reference is inappropriate. LM2009 do show a split scheme but it is not really implicit. I find (8) to (9) vague. Too convoluted. Simply give the entire scheme in 9 (there is ample space) then we all know what is done rather then having to piece somehow together what is factually done (which I failed to do). Comment whether the schemes for Sf and rain are 2nd or 1st order also per the comment on eqn (8) below. Given that Sf is kind of quadratic in u (or hu) a semi-implicit scheme is almost instantly made up. That would be straightforward for (9a) and even easy for (9b). One can check the formal time accuracy for the case with only Sf and R on the RHS. And perhaps even for a constant bottom slope river kinematic limit (which is an exact limit within the SWE).

Line 121b: Why is DG2 slower; due to the CFL=1/3 criterion relative to CFL ∼0.9 for FV1?
 Eqn (8): why is this scheme for rainfall 2nd order? Why is rain not directly included into (9). R=R(t) so non-autonomous RK would be fine? Please explain your choices better. Please provide the entire scheme for the entire system in (9); there is space and currently the time stepping scheme is unclear. Please clarify.

Figure 2: How can one apply rainfall if dt is not know yet, since dt is determined later in the diagram? Order seems off?

Line 157: typo; L missing (only subscript y seen).

Line 173: add OUP "and," at end of third option.

Line 183: Can it be clarified earlier whether or not (I think it is "or not") DG2 can be truly C^0? Since the x*y term is not include, therefor it cannot represent C^0 solutions or topography. While speed is gained this drops formal C^0 smoothness. Say so explicitly,

probably earlier, also why this option with only polynomials 1, x and y is chosen, I assume to enhance computational speed.

Line 186: Nothing is said about the time stepper for FV1 and ACC? Why does FV1 do better for thin overland flows? How are Sf and R dealt with in FV1? This information is relevant to understand the test results provided and analysed later. Please clarify (earlier on).

Eqn (14): Perhaps add some spacing around \Delta t.

Line 207: Second-order wrt to what since it cannot be 2nd order wrt solutions requiring the advective velocity terms? Please define clearly what is meant here? When advection is important, it cannot be second order, of course?

Fig. 6: Since FV1 is order one that use of the same grid spacing in the different models is a priori unfair. What happens if FV1 is adjusted (and maybe ACC) such that the expected errors are the same? Please add that situation. The left bottom panel could be done for finer resolution as well, as in Fig 7.

Fig. 7: Figure a bit large. Otherwise nice.

Line 270 paragraph: But this comparison is unfair; one needs to compare run times for the same or at least similar accuracy; either focussing on overall accuracy or the accuracy at the front since the latter may be most relevant. This needs an extension.

Line 283: Statement by authors themselves underscores my previous remark. So why is this case not made in the paper with the same convergence/accuracy compared with runtime?

Fig. 8b,c: Same remark; what is the right comparison here? Please address. Caption: page number and caption overlap.

Fig. 12: Can you show simulations for DG2 with dx=50m, say, as well since that runs in less time than the other models but the solution may still be better?

Table 1: So DG2 dx=50m should be compared with runtimes of the other models or even DG2-GPU at dx=25m should be compared with the other models, in line with the previous five remarks.

Line 352: Why is the domain initially dry? How long does it take for the memory of this odd initial condition to disappear?

Line 356: I do not understand this remark "No Data"; translate this computer remark.

Lin 364: But that seems unfair. Finer data need to be projected on the topography that DG2 can model, which is not dx=40m but the finer one projected? In fact, DG2 would then do a bit better I suspect. Address this please; perhaps really redo the DG2 case; it should have 3 dofs per cell, also for the topography.

Line 367: Why is repositioning required since across the river channel the river level is within 0.2m, say, the same. Is the level meant the river level height above sea level? What is recorded? So why is this repositioning needed, since variation across the river channel will be minimal. It is free-surface height that matters. Please clarify.

Fig. 14: I am not convinced by the DG2 results due to not taking into account slope information in the bathymetry. So DG2 may be underperforming while this can partially be repaired. Please update and clarify.

Line 408: As stated above this anomaly of using only a planar representation in DG2 should be fixed: you do not use DG2 optimally with only 40m planar resolution on bathymetry, while you are allowed to include higher resolution and project on the DG2 basis you use. This leads to an unfair comparison which seems not aligned with the DG2 capabilities.

Line 418: What is the error in the observations? What is the variation in surface level across a (flooded) river channel? What is consequently the result of shifting the measurement position? I would guess the error is circa 0.2m (from watching river levels bob about +-0.1m for a river in flood next to a river gauge). What is the variation within
10m along stream of the surface level in the simulation and across the channel, in order to get an error estimate? Please clarify and add a discussion on errors, in both the observations and simulations.

Line 425: Please add the word "maximum" before the last word "flood".

Line 428: "More notable differences" Where? Clarify please, e.g. by using arrows.

Line 428: On flood defence walls not being captured on coarse grids: unless, of course, you put those in by hand, which should be done (regardless). Here is where variable/non-uniform mesh capabilities of DG2 or FV1 would have come in handy. These are then essentially (vertical) walls with a finite vertical extent. Also on a coarse mesh one can add vertical walls at cell edges as approximation.

Line 432: So, what then is right or wrong, ACC or FV1 and how do we know that? Please clarify.

Fig. 16: Graphs in Fig. 16 look all the same; use arrows to highlight areas you want the reader to focus on.

Fig. 17: I still disagree with a DG0 bottom topography at dx=40 for DG2. That is a false comparison, cf. previous remarks made. Further: what flood accuracy do we actually care about and what are the error bars given errors in rainfall? Please discuss.

Line 491: But as said before, your topography is incorrect, in not matching DG2 accuracy that can be used. Also, I am not convinced by your time stepping in DG2 as compared to what is done in ACC and FV1 (the latter which is unclear as well and makes it difficult to understand the comparison). Please clarify.

Line 501: The conclusion here is a bit of a downer for DG2; no discussion is made on using hybrid approaches, including variables time and spatial resolution for which DG should be good and more promising. Please comment and update.

Appendix B: I am a bit confused; why do we need to linearise to establish whether ACC

is second order? Please clarify.

---

## Author Comment (AC1) · 4 Apr 2021

Dear Editor and Referee #1,

We thank you for the time spent to evaluate and review our manuscript and for your positive recommendations.

Referee #1 raises a number of minor comments, which were all addressed in the revised manuscript. Point-by-point answers to the Referee #1 are provided below, whereby we explain how and where each point raised was incorporated into the revised manuscript.

Answers are in blue text and appear after each reviewer comment, followed by

a box showing the associated changes applied to the revised manuscript.

Sincerely,

Georges Kesserwani and James Shaw
* * *
"Congratulations to this very clear and concise piece of work. The overall goal, methods, and results of the model development and model application are described with sufficient detail, and the overall structure of the paper supports reading and understanding of the content. Also, the presentation of the results is in most cases clear and informative."
We thank the reviewer for their interest in our paper, and we are glad the manuscript was clear and concise. The manuscript has been revised to address all the referee's comments, as we clarify in the following.

"The fact that model code and study results are provided on Zenodo is very much appreciated, great! A short readme or similar would help, however, to guide the user through the folder structure of the repository."
The Zenodo record description at https://zenodo.org/record/4066824 has been substantially expanded to document the folder structure, file format and filename conventions. Also, the website, https://www.seamlesswave.com/LISFLOOD8.0, was cited in the abstract and introduction. This website allows readers to access step-by-step documentation of how to run LISFOOD-FP8.0 over a series of test cases.

"1. Page 2 / lines 52-43: maybe good to mention the license under which the model is distributed? This gives directly an idea of what is permitted and not. Additionally, I strongly encourage you to add a DOI in the manuscript linking to the Zenodo-version used in this manuscript."
The abstract and the introduction section has been revised to mention the license under which the model is distributed.

> **Abstract**
>
> LISFLOOD-FP 8.0 therefore marks a new step towards operational DG2 flood inundation modelling at the catchment scale. LISFLOOD-FP 8.0 is freely available under the GPL v3 license, with additional documentation and case studies at https://www.seamlesswave.com/LISFLOOD8.0.

> **1 Introduction**
>
> This paper presents a new LISFLOOD-DG2 solver of the full shallow water equations, which is integrated into LISFLOOD-FP 8.0 and freely available  under the GNU GPL v3 license (LISFLOOD-FP developers, 2020).

"2. Page 11 / line 213: why are these recent optimizations lacking? Please provide a brief explanation."
The optimised LISFLOOD-ACC solver specified by Neal et al. (2018) implements a sub-grid channel model (Neal et al., 2012a) and CPU-specific optimisations that do not translate naturally to GPU architectures, and this optimised ACC solver does not yet support the rain-on-grid features used later in Sect. 3.3. To

facilitate an intercomparison between solvers, the LISFLOOD-ACC solver used here is the version specified by Neal et al., (2012b), which supports the necessary rain-on-grid features and shares the same algorithmic approach as the FV1 and DG2 solvers.
* * *
**3 Numerical results**

The optimised LISFLOOD-ACC solver  specified by Neal et al., 2018 implements a sub-grid channel model (Neal et al., 2012a) and CPU-specific optimisations that do not translate naturally to GPU architectures. Additionally, at the time that model runs were performed, the optimised ACC solver did not yet support the rain-on-grid features used later in Sect. 3.3[a]. To facilitate a like-for-like intercomparison between solvers, the LISFLOOD-ACC solver used here is the version specified by Neal et al., 2012b, which already supports the necessary rain-on-grid features and shares the same algorithmic approach as the FV1 and DG2 solvers.
* * *
[a]Rain-on-grid features have since been added to the optimised ACC solver, and will be available in a future LISFLOOD-FP release.
* * *
**3.1.3 Solver runtimes for a varying number of elements**

To facilitate a like-for-like comparison with FV1 and DG2, ACC solver runtimes were obtained for the ACC implementation of Neal et al., 2012a.
...
It is expected that additional performance can be gained by using the alternative, CPU-optimised ACC implementation (Neal et al., 2018), and these CPU-specific optimisations are also under consideration for future enhancement of the DG2 and FV1 solvers.
* * *
"3. Section 3.1 and 3.2: for both test cases, it is stated that they are widely used to assess model predictions. What I thus would like to see is an extended discussion of how model results here generally compare to other modeling studies. For instance, are the trend presented here and their magnitude comparable? A bit more depth is needed to move the manuscript more towards a scientific publication rather than an extended model documentation logbook."

In section 3.1.1, the previous version of the manuscript already mentioned that "DG2, FV1 and ACC predictions of water depth and velocity agree closely with existing industrial model results (Fig. 4.10 and 4.11 in Néelz and Pender (2013)).". A sentence was added to further discuss the modelling studies in (Huxley et al. 2017, Table 11)
* * *
**3.1.1 Water depth and velocity hydrographs**

Differences in velocity predictions are more pronounced (Fig. 6e–h). The biggest differences are seen at point 1 (Fig. 6e), located only 50 m from the breach, since the flow at this point is dominated by strong inflow discharge with negligible retardation by frictional forces. At point 1, ACC and DG2 velocity predictions agree closely with the majority of industrial models (Fig. 4.11 in Néelz and Pender (2013) ). LISFLOOD-FV1 predicts faster velocities up to $0.5$ ms$^{-1}$, which is close to the prediction of TUFLOW FV1 (Huxley et al., 2017, Table 11).
* * *
For section 3.2.1, two sentences have been added to further discuss the modelling studies in (Huxley et al. 2017) for TUFLOW solvers and the (Ayog et al. 2021) including results from a MUSCL-FV2 solver at 10 m resolution.

**3.2.1 Water level and velocity hydrographs**

Predicted water level and velocity hydrographs are shown in Fig. 11. The water level hydrographs show that water ponds in small topographic depressions at point 1 (Fig. 11a), point 3 (Fig. 11b) and point 5 (Fig. 11c). Point 7 is positioned near the steep valley slope, and is only inundated between $t = 1$ hour and $t = 8$ hours (Fig. 11d). At both resolutions, water levels predicted by all solvers agree closely with existing industrial model results at points 1, 3 and 7 (Fig. 4.16 in Néelz and Pender, 2013). Small water level differences accumulate as water flows downstream and at point 5, positioned farthest downstream of the dam break, differences of about 0.5 m are found depending on the choice of resolution and solver (Fig. 11c). Similar water level differences have been found amongst the suite of TUFLOW solvers (Huxley et al., 2017) and amongst other industrial models (Néelz and Pender, 2013).
Bigger differences are found in velocity predictions, particularly at locations farther downstream at point 3 (Fig. 11f), point 4 (Fig. 11g) and point 7 (Fig. 11h). At point 3, DG2 predicts small, transient velocity variations at $\Delta x = 50$ m starting at $t = 1$ hour; these variations are not captured by the FV1 or ACC solvers, but have been captured by a FV2-MUSCL solver at the finest resolution of $\Delta x = 10$ m, as reported by Ayog et al., 2021.

"4. Chapter 3.1.2: Here, the cases are benchmarked for two different resolutions. The initial 5 m resolution and a finer 1 m resolution. Question 1 is, how did you derive the new geographical data, how did you perform the resampling? And question 2, why did you not look into grid coarsening - wouldn't the chance of having abrupt water level changes between cells become larger when applying grids with, admittedly probably very much, coarser spatial resolution? This would be less relevant to assess speed of the solvers, but accuracy."
Reply to Question 1: Spatial grid convergence is studied by modelling at grid resolutions between $\Delta x = 5$ m and $\Delta x = 1$ m. Since the floodplain is flat, no topographic resampling is required.

**3.1.2 Spatial grid convergence**

Spatial grid convergence is studied by modelling at grid resolutions of $\Delta x = 5$ m, 1 m, and 0.5 m. Since the floodplain is flat, no topographic resampling is required. On each grid, the water depth cross-section is measured along the centre of the domain (Fig 7). DG2, FV1 and ACC cross-sectional profiles at the standard grid spacing of $\Delta x = 5$ m agree well with industrial model results (Fig 4.13 in Néelz and Pender (2013)).

Reply to Question 2: Coarsening beyond $\Delta x = 5$ m yielded a similar conclusion to that reached by the results at $\Delta x = 5$ m, so additional coarse results were excluded to ensure the clarity of Fig. 7.

"5. Chapter 3.3: While for the two Environment Agency test cases a motivation was stated, it does not become clear where why exactly this case study was selected. Please elaborate briefly why you decided to use this test case and not another one from the rich literature of LISFLOOD-FP studies, for example."
The Storm Desmond case study was selected for its spatially- and temporally-varying rainfall across a large catchment. This dynamic rain-on-grid capability was unavailable in earlier versions of LISFLOOD-FP and therefore unavailable in earlier LISFLOOD-FP studies.

> ### 3.3 Catchment-scale rain-on-grid simulation
>
> In December 2015, Storm Desmond caused extensive fluvial flooding across the Eden catchment in North West England (Szönyi et al., 2016). This storm event has previously been simulated using a first-order finite volume hydrodynamic model (Xia et al., 2019), with  overland flow and fluvial flooding driven entirely by  spatially- and temporally-varying rainfall data over the 2500 km$^2$ catchment. As such, this simulation is ideally-suited to assess the  new rain-on-grid capabilities in LISFLOOD-FP 8.0, and represents one of the first DG2 hydrodynamic modelling studies of rainfall-induced overland flow  across a large catchment.

"6. Page 27 / Line 423: this is a very important aspect and should be highlighted more prominently in the manuscript (e.g. abstract and/or summary). One of the key reasons many scholars/practitioners use LISFLOOD-FP is its sub-grid scheme. This also holds for the comment made in point 2."

The revised introduction clarifies that, "Since the new DG2 and FV1 solvers are purely two-dimensional and parallelised for multi-core CPU and GPU architectures, the new solvers do not currently integrate with the LISFLOOD-FP sub-grid channel model (Neal et al., 2012a) or incorporate the CPU-specific optimisations available to the ACC solver(Neal et al., 2018)."

> ### 1 Introduction
>
>  Since the new DG2 and FV1 solvers are purely two-dimensional and parallelised for multi-core CPU and GPU architectures, the new solvers do not currently integrate with the LISFLOOD-FP sub-grid channel model (Neal et al., 2012a) or incorporate the CPU-specific optimisations available to the ACC solver (Neal et al., 2018).

The revised opening of Sect. 3 also clarifies that, "The optimised LISFLOOD-ACC solver specified by Neal et al., 2018 implements a sub-grid channel model (Neal et al., 2012a) and CPU-specific optimisations that do not translate naturally to GPU architectures." (recall the changes made in response to point 2 above).

> ### Abstract
>
> The solvers are parallelised on multi-core CPU and Nvidia GPU architectures and run existing LISFLOOD-FP modelling scenarios without modification. These new, fully two-dimensional solvers are available alongside the existing local inertia solver (called ACC), which is optimised for multi-core CPUs and integrates with the LISFLOOD-FP sub-grid channel model.

"7. Chapter 3.3.2: For the analyses of flood extent, would it not be useful to include metrics like the hit rate, false alarm ration and critical success index to quantify the actual (dis)agreement between simulations and results?"

We followed the reviewer recommendations and investigated the hit rate, false alarm and critical success index metrics, given their relevance to quantify the analysis of floodplain comparisons.

**Table 3**

Hit rate (H), false alarm ratio (F) and critical success index (C) for the DG2 and FV1 predictions of maximum flood extent calculated against the reference solution of the ACC solver at $\Delta x = 10$ m.

| | $\Delta x = 40$ m | | | $\Delta x = 20$ m | | | $\Delta x = 10$ m | | |
|------|------|------|------|------|------|------|------|------|------|
| | H | F | C | H | F | C | H | F | C |
| DG2 | 0.83 | 0.32 | 0.59 | 0.86 | 0.55 | 0.40 | 0.85 | 0.56 | 0.40 |
| FV1 | 0.77 | 0.27 | 0.59 | 0.80 | 0.19 | 0.67 | 0.93 | 0.03 | 0.90 |

**3.3.2 Maximum flood extent over Carlisle**

The DG2 and FV1 predictions of maximum flood extent can be quantified against the ACC prediction at $\Delta x = 10$ m, which is treated as the reference solution. The hit rate measures flood extent underprediction as the proportion of wet elements in the reference solution that were also predicted as wet. The false alarm ratio measures flood extent overprediction as the proportion of predicted wet elements that were dry in the reference solution. The critical success index measures both over- and underprediction. All three metrics range between 0 and 1, and further details are provided by Wing et al., 2017.
The hit rate (H), false alarm ratio (F) and critical success index (C) are given in Table 3. At $\Delta x = 40$ m, the critical success index is 0.59 for both DG2 and FV1, but DG2 has a higher hit rate and false alarm ratio, suggesting that DG2 predicts a wider flood extent than ACC or FV1. At $\Delta x = 20$ m and $\Delta x = 10$ m, the false alarm ratio and critical success index for DG2 deteriorate, but a hit rate of 0.83–0.86 is maintained, which is acceptable given that high-resolution predictions are downscaled from the DG2 piecewise-planar solution at $\Delta x = 40$ m. FV1 predictions at $\Delta x = 20$ m and $\Delta x = 10$ m are obtained directly without downscaling, and FV1 predictions converge towards ACC predictions with successive grid refinement. This convergence is evidenced in all three metrics, with FV1 at $\Delta x = 10$ m achieving a high hit rate (0.93), low false alarm ratio (0.03), and high critical success index (0.90).

"8. Figure 16 and Figure 17: This figure is hard to read. While adding the OSM background map is appreciated for geographical reference, (as done in a figure above), it's diluting the actual information about the flood maps in the current form. Please consider revising this figure." The OpenStreetMap background was removed from Figures 16 and 17.

"9. Summary/Conclusions: this section nicely wraps up the manuscript. However, recommendations for further studies and improvements, and the shortcomings of the current version (both feature-wise and technologically), are missing. Please set your work in context of what was done so far, how your work adds to that and opens up new possibilities, and what challenges are still lying ahead to fast hydrodynamic simulation over coarse and large domains"
The challenges that lie ahead to improve DG2 flood modelling over coarse and large domains as well as the possibilities for future improvements have been discussed in the revised "Summary and conclusions" section.

**4 Summary and conclusions**

However, FV1 and DG2 are the first solvers in LISFLOOD-FP to gain a dynamic rain-on-grid capability, with this capability being added to the optimised ACC solver in a future release. To further improve efficiency and accuracy at coarse resolutions over large catchments, one future direction would be to port the sub-grid channel model—currently integrated with the CPU-optimised ACC solver—to GPU architectures. Another useful direction would be to enable a multi-resolution solver based on Kesserwani and Sharifian, 2020, and introduce a hybrid DG2/FV1 solver that downgraded the DG2 formulation to FV1 in regions of very thin water layer, or in regions of finest grid resolution, to further reduce the computational cost. Both directions are being investigated for inclusion in future LISFLOOD-FP releases.

---

## Author Comment (AC2) · 4 Apr 2021

Dear Editor and Referee #2,

We thank you for the time spent to evaluate and review our manuscript and for the supportive recommendations.

The major and minor remarks brought by Referee #2 were all addressed in the revised manuscript. In our point-by-point answers below, we explain how and where each point raised was addressed in the revised manuscript.

Answers are in blue text and appear after each reviewer comment, followed by

a box showing the associated changes applied to the revised manuscript.

Sincerely,

Georges Kesserwani and James Shaw
* * *
**Major remarks**

"The inclusion of a second-order DG2 discretisation and implementation in the LISFLOOD-FP 8.0 flood forecasting suite, on a regular mesh, of the shallow water equations with friction and rainfall, for flood forecasting, is presented, including analyses of three test cases and comparison with a local inertia solver ACC and a first-order FV1 discretisation and implementation. Parallelisation and GPU results are intercompared in terms of performance and runtime. While the paper is very interesting and the work constitutes a useful addition in a flood forecasting suite for public use, which is very important, there seem to be several loose ends and unclear aspects that need to be resolved before publication is warranted. I therefore recommend to return the manuscript for major revisions."
Thank you for highlighting the interest and importance of this paper, and for the useful comments and recommendations. They were very helpful to clarify the presentation of the DG2 solver and the discussions of its performance.

"It would be desirable that the following major issues are addressed, either clarified or resolved:"
Explanations on how these issues were resolved or clarified are explained below each comment.

"(i) Since ACC, FV1 and DG2 schemes are tested and compared more clarity on their characteristics is desired in order to understand the differences in the results presented. Why is DG2 much slower than FV1 in the third test? The time stepping schemes of FV1 and DG2 are unclear. Figuring out the split scheme for DG2 given rainfall and damping is too hard or impossible also, since the references are unclear; it is easier to simply state the full scheme; there is sufficient space in (9) to do so and remove any ambiguity. The split scheme for Sf is not implicit, once one looks into the references. Why is the handling of friction (Sf) in DG2 leading to troubles for thin, fast overland flow while it is not troubling for FV1 and ACC? Perhaps use the same time stepping scheme in DG2 as for FV1 or make aspects of Sf semi-implicit. Please clarify and make improvements"
Section 2, mostly subsection 2.1, has been substantially revised to include the technical details of the DG2 and FV1 solvers including their time stepping schemes for the splitting

friction integration and the discretisation of the rainfall source term. In the new subsection 2.1.2, the split friction scheme with DG2/FV1 is explicitly presented. The subsection also includes an explanation on why the handling of friction with DG2 can reduce the time-step size for thin, fast overland flow compared to FV1.

**2.1.2 Discretisation of the friction source term**

The discretisation of the friction source term is  based on the split implicit scheme  of Liang and Marche, 2009. Without numerical stabilisation, the friction function $C_f = g n_M^2 / h^{1/3}$ can grow exponentially as the water depth vanishes at a wet-dry front, but the scheme adopted here is designed to ensure numerical stability by limiting the frictional force to prevent unphysical flow reversal.
The implicit friction scheme is solved directly (see Sect. 3.4, Liang and Marche 2009) such that frictional forces are applied to the $x$-directional discharge component $q_x$ over a time-step $\Delta t$, yielding a retarded discharge component $q_{fx}$:

$$q_{fx}(\mathbf{U}) = q_x + \Delta t \frac{S_{fx}}{\mathcal{D}_x}, \tag{11a}$$

where the denominator $\mathcal{D}_x$ is

$$\mathcal{D}_x = 1 + \left( \frac{\Delta t C_f}{h} \right) \left( \frac{2u^2 + v^2}{\sqrt{u^2 + v^2}} \right). \tag{11b}$$

To update the element-average discharge coefficient $q_{x_{i,j,0}}$, Eqn. (11) is evaluated at the element centre :

$$q_{x_{i,j,0}}^{n+1} = q_{fx}(\mathbf{U}_{i,j,0}^n), \tag{12a}$$

while the slope coefficients $q_{x_{i,j,1x}}$ and $q_{x_{i,j,1y}}$ are updated by calculating the $x$- and $y$-gradients using evaluations of Eqn. (11) at Gaussian quadrature points Gx1, Gx2, and Gy1, Gy2 (Fig. 1):

$$q_{x_{i,j,1x}}^{n+1} = \frac{1}{2} \left[ q_{fx}(\mathbf{U}_{i,j}^{\mathrm{Gx2}}) - q_{fx}(\mathbf{U}_{i,j}^{\mathrm{Gx1}}) \right], \tag{12b}$$

$$q_{x_{i,j,1y}}^{n+1} = \frac{1}{2} \left[ q_{fx}(\mathbf{U}_{i,j}^{\mathrm{Gy2}}) - q_{fx}(\mathbf{U}_{i,j}^{\mathrm{Gy1}}) \right]. \tag{12c}$$

Similarly, frictional forces are applied to the $y$-directional discharge component $q_y$ yielding a retarded discharge $q_{fy}$:

$$q_{fy}(\mathbf{U}) = q_y + \Delta t \frac{S_{fy}}{\mathcal{D}_y}, \tag{13a}$$

$$\mathcal{D}_y = 1 + \left( \frac{\Delta t C_f}{h} \right) \left( \frac{u^2 + 2v^2}{\sqrt{u^2 + v^2}} \right). \tag{13b}$$

> While this friction scheme has been successfully adopted in finite-volume and discontinuous Galerkin settings for modelling dam break flows and urban flood events (Want et al., 2011; Kesserwani and Wang, 2014), it can exhibit spuriously large velocities and correspondingly small time-steps for large-scale, rainfall-induced overland flows, involving widespread, very thin water layers flowing down hill slopes and over steep river banks, as demonstrated by Xia et al., 2017. Due to the involvement of the slope coefficients, water depths at Gaussian quadrature points can be much smaller (and velocities much larger) than the element-average values. Therefore, for overland flow simulations, the LISFLOOD-DG2 time-step size is expected to be substantially reduced compared to LISFLOOD-FV1, which only involves element-average values.

"(ii) In test 3, but also in earlier tests, DG2 has topography at dx=40 (or another dx for tests 1 and 2) and seemingly the higher degrees of freedom used for the variables is not used for the topography, i.e. the dx=10 topography information can be used to make finer projections onto the topography given that DG2 is second order and the topography should not be planar. Hence, the (at times and in certain simulations) observed simulation underperformance of DG2 seems to be/is more severe than needed."

It is important to clarify that the topography with the proposed DG2 solver has been, and must be, projected using three degrees of freedom (Kesserwani et al. 2018, Kesserwani and Sharifian 2020, Ayog et al. 2021). This is what we referred to as "piecewise-planar" topography as its projection is expanded based on three degrees of freedom reconstructed from four DEM data evaluation at the four vertices (as opposed to with ACC/FV1 that uses "piecewise-constant" topography, extracted from a single evaluation from the DEM data). To avoid this misunderstanding, Section 2, mostly subsection 2.1 and figure 1, has been revised to describe how the DG2 topography projection was generated for the DG2 solver based on three degrees of freedom.

**2.1.1 Initialisation of piecewise-planar topography coefficients from a DEM raster file**

The topography coefficients $[z_{i,j,0}, z_{i,j,1x}, z_{i,j,1y}]$ are initialised to ensure the resulting piecewise-planar topography is continuous at face centres, where Riemann fluxes are calculated and the wetting-and-drying treatment is applied under the well-balancedness property (Kesserwani et al., 2018). The topographic elevations at the N, S, E, and W face centres are calculated by averaging the DEM raster values taken at the NW, NE, SW and SE vertices (Fig. 1) such that $z_{i,j}^{\mathrm{N}} = (z_{i,j}^{\mathrm{NW}} + z_{i,j}^{\mathrm{NE}})/2$ and similarly for $z_{i,j}^{\mathrm{E}}$, $z_{i,j}^{\mathrm{S}}$, and $z_{i,j}^{\mathrm{W}}$. The element-average coefficient $z_{i,j,0}$ is then calculated as:

$$z_{i,j,0} = \frac{1}{4}\left[z_{i,j}^{\mathrm{NW}} + z_{i,j}^{\mathrm{SW}} + z_{i,j}^{\mathrm{NE}} + z_{i,j}^{\mathrm{SE}}\right], \tag{14a}$$

while the slope coefficients $z_{i,j,1x}$ and $z_{i,j,1y}$ are calculated as the gradients across opposing face centres:

$$z_{i,j,1x} = \frac{1}{2\sqrt{3}}\left(z_{i,j}^{\mathrm{E}} - z_{i,j}^{\mathrm{W}}\right), \tag{14b}$$

$$z_{i,j,1y} = \frac{1}{2\sqrt{3}}\left(z_{i,j}^{\mathrm{N}} - z_{i,j}^{\mathrm{S}}\right). \tag{14c}$$

LISFLOOD-FP 8.0 includes a utility application, `generateDG2DEM`, that loads an existing DEM raster file and outputs new raster files containing the element-average, $x$-slope and $y$-slope topography coefficients, ready to be loaded by the LISFLOOD-DG2 solver.

"(iii) DG2 is used on a regular mesh; DG is most optimal for non-uniform situations but this option or restriction is not discussed; also not discussed are hybrid 1D and 2D schemes or hybrid FV1 and DG2 options, the latter which should be easy to accommodate. At least a discussion is warranted."

Discussions have been added in the revised paper to highlight potential alternatives to DG2 on uniform grids, including the consideration of a non-uniform DG2 solver, linkage with the existing sub-grid channel model on LISFLOOD-FP and a hybrid FV1/DG2.

**3.3.1 River channel free-surface elevation hydrographs**

Spatially-adaptive solvers (Kesserwani and Sharifian, 2020; Özgen-Xian et al., 2020) and non-uniform meshing techniques (Kolega and Syme, 2019) offer another alternative to improve flow predictions by selectively capturing fine-scale channel geometries, and such methods are under development for inclusion in a future LISFLOOD-FP release.

**3.3.2 Maximum flood extent over Carlisle**

The representation of these flood defences could be improved by adopting the recently-developed LISFLOOD-FP levee module[a] (Wing et al., 2019; Shustikova et al., 2020), or by implementing a spatially-adaptive multi-resolution method that selectively refines the grid resolution around river channels and other fine-scale features (Kesserwani and Sharifian, 2020).
* * *
[a]Not yet available with the FV1 or DG2 solvers.

**4 Summary and conclusions**

However, FV1 and DG2 are the first solvers in LISFLOOD-FP to gain a dynamic rain-on-grid capability, with this capability being added to the optimised ACC solver in a future release. To further improve efficiency and accuracy at coarse resolutions over large catchments, one future direction would be to port the sub-grid channel model—currently integrated with the CPU-optimised ACC solver—to GPU architectures. Another useful direction would be to enable a multi-resolution solver based on Kesserwani and Sharifian, 2020, and introduce a hybrid DG2/FV1 solver that downgraded the DG2 formulation to FV1 in regions of very thin water layer, or in regions of finest grid resolution, to further reduce the computational cost. Both directions are being investigated for inclusion in future LISFLOOD-FP releases.

"(iv) Some of the speed and/or accuracy comparisons seem incomplete or unfair: e.g., in Fig. 6, not the same resolution should (only) be compared but (also) the speed for mixed resolutions with (roughly) the same accuracy"

Test 1 and Test 2 (Subsection 3.1 and 3.2) have been extended to also compare speed-ups among the solvers for mixed resolutions leading to outputs with (roughly) the same accuracy (see revised Fig. 7, revised Fig. 12, revised Table 1, and the changes made listed below).

**3.1.3 Solver runtimes for a varying number of elements**

As seen earlier in the inset panel of Fig. 7, similar wave-fronts were predicted by DG2 at $\Delta x = 5$ m, ACC at $\Delta x = 2$ m, and FV1 at $\Delta x = 0.5$ m. At these resolutions, DG2-CPU, DG2-GPU and ACC achieved a similar solution quality for a similar runtime cost, with all solvers completing in about 4 minutes (Fig. 8a). Meanwhile, the DG2 solvers on a ten-times coarser grid were 140× faster than FV1-CPU (10 hours 42 minutes) and 28× faster than FV1-GPU (1 hour 47 minutes).

**3.1.4 Multi-core CPU scalability**

The FV1, ACC and DG2 solvers converged on similar water depth solutions with successive grid refinement. Owing to its first-order accuracy, FV1 requires a very fine resolution grid to match the solution quality of DG2 or ACC, though FV1-GPU enables runtimes up to $6\times$  faster than the 16-core FV1-CPU solver. Thanks to its second-order accuracy, DG2 water depth predictions are spatially converged at coarser resolutions (Fig. 7).  Hence, DG2  is able to replicate the modelling quality of FV1 at a much coarser resolution, and the multi-core DG2-CPU solver is  a competitive choice for grids with fewer than 100,000   elements.

**Table 1**

Solver runtimes at  grid spacings of $\Delta x = 50$ m, $\Delta x = 20$ m, and  $\Delta x = 10$ m. ACC, FV1-CPU and DG2-CPU solvers are run on a 16-core CPU; FV1-GPU and DG2-GPU solvers are run on a single GPU. ACC solver runtimes were obtained for the ACC implementation of Neal et al., 2012a

|  | $\Delta x = 50$ m
 57 000 elements | $\Delta x = 20$ m
850 000 elements | $\Delta x = 10$ m
1.7 million elements |
|---|---|---|---|
| ACC | 20 s | 466 s (8 mins) | 1779 s (30 mins) |
| FV1-CPU | 22 s | 739 s (12 mins) | 2188 s (36 mins) |
| FV1-GPU | 19 s | 145 s (2 mins) | 965 s (16 mins) |
| DG2-CPU | 788 s (13 mins) | 4133 s (69 mins) | 33009 s (9 hours) |
| DG2-GPU | 448 s (7 mins) | 2304 s (38 mins) | 13606 s (4 hours) |

> ### 3.2.3 Runtime cost
>
> DG2-CPU  at $\Delta x = 10$ m, while still achieving similarly accurate flood map predictions at a $5\times$  coarser resolution (Fig. 12). DG2-CPU at  $\Delta x = 50$ m is $2\times$ faster than ACC at $\Delta x = 10$ m, while DG2-GPU is twice as fast again. DG2-GPU flood maps at an improved resolution of $\Delta x = 20$ m are obtained at a runtime cost of 38 mins, which is still competitive with ACC at $\Delta x = 10$ m (with a runtime cost of 30 mins).
> In summary, all solvers predicted similar water depth and velocity hydrographs, though ACC experienced a short period of numerical instability in a localised region where the Froude number exceeded the limit of the local inertia equations. The shock-capturing FV1 and DG2 shallow water solvers yield robust predictions throughout the entire simulation. with FV1-GPU being consistently faster than ACC on a 16-core CPU. As found earlier in Sect. 3.1.3, DG2 at a 2–5$\times$ coarser resolution is a competitive alternative to ACC and FV1, with the GPU implementation being preferable when running DG2 on a grid with more than 100,000 elements.

"(v) The is a large accumulation of minor remarks, see below, which should be refuted and addressed. The above major remarks are also reflected, sometimes on several occasions, in several of the minor/detailed remarks given below."
Description of how the minor/detailed remarks have been addressed is provided below.

**Minor/detailed remarks**

"Line 23: simplify (plural)"
Revised to resolve any ambiguity in ACC and ATS simplifications.

> ### 1 Introduction
>
> LISFLOOD-FP already includes a  local inertia (or 'gravity wave') solver, LISFLOOD-ACC, and a  diffusive wave (or 'zero-inertia') solver, LISFLOOD-ATS. The LISFLOOD-ACC  solver simplifies the full shallow water equations by neglecting convective acceleration, while LISFLOOD-ATS neglects both convective and inertial acceleration.

"Line 42: But there are plenty of FV2 solvers, which may be faster; has a comparison with those been made or at least discussed? Please discuss."
A discussion has been added to the introduction section.

> ### 1 Introduction
>
> Second-order finite volume (FV2) methods offer an alternative approach to obtain second-order accuracy, with many FV2 models adopting the Monotonic Upstream-centred Scheme for Conservation Laws (MUSCL) method. While FV2-MUSCL solvers can achieve second-order convergence (Kesserwani and Wang, 2014), the MUSCL method relies on global slope limiting and non-local, linear reconstructions across neighbouring elements that can affect energy conservation properties (Ayog et al., 2021) and affect wave arrival times when the grid is too coarse (Kesserwani and Wang, 2014). Hence, although FV2-MUSCL is typically 2–10× faster than DG2 per element (Ayog et al., 2021), DG2 can improve accuracy and conservation properties on coarse grids, which is particularly desirable for efficient, long-duration continental- or global-scale simulations that rely on DEM products derived from satellite data (Bates 2012, Yamazaki et al., 2019).

"Line 56: What about mixed 1D and 2D solvers, see the literature?"
On this aspect, a sentence has been added to only discuss the literature specific to LISFLOOD-FP.

> ### 1 Introduction
>
> LISFLOOD-FP includes extension modules to provide efficient rainfall routing Sampson et al., 2013, modelling of hydraulic structures (Wing et al., 2019; Shustikova et al., 2020), and coupling between two-dimensional flood-plain solvers and a one-dimensional sub-grid channel model (Neal et al., 2021a)

"Line 58: This paper [use of "remainder" is a bit much on page 2]." Amended.

> ### 1 Introduction
>
> The  paper is structured as follows: ...

"Line 72: What is new or simplified relative to K2018, and why?" Addressed by adding the following introductory text at the beginning of Sect. 2.1.

> **2.1 The new LISFLOOD-DG2 solver**
>
> The LISFLOOD-DG2 solver implements the DG2 formulation of Kesserwani et al., 2018 that adopts a simplified 'slope-decoupled' stencil compatible with raster-based Godunov-type finite volume solvers. Piecewise-planar topography, water depth and discharge fields are modelled by an element-average coefficient and dimensionally-independent $x$-slope and $y$-slope coefficients. This DG2 formulation achieves well-balancedness for all discharge coefficients in the presence of irregular, piecewise-planar topography with wetting-and-drying (Kesserwani et al., 2018) . A piecewise-planar treatment of the friction term is applied to all discharge coefficients prior to each time-step, based on the split implicit friction scheme of Liang and Marche, 2009. Informed by the findings of Ayog et al., 2021, the automatic local slope limiter option in LISFLOOD-DG2 is deactivated for the flood-like test cases presented in Sect. 3. This slope-decoupled, no-limiter approach can achieve a 5× speed-up over a standard tensor-product stencil with local slope limiting (Kesserwani et al., 2018; Ayog et al., 2021), meaning this DG2 formulation is expected to be particularly efficient for flood modelling applications.

"Please specify Eqn (2): define x, y, t."
These are now defined after they first appear in the partial derivatives in Eqn (1):

> **2.1 The new LISFLOOD-DG2 solver**
>
> The DG2 formulation (Kesserwani et al., 2018) discretises the two-dimensional shallow water equations, written in conservative vectorial form as
>
> $$\partial_t \mathbf{U} + \partial_x \mathbf{F}(\mathbf{U}) + \partial_y \mathbf{G}(\mathbf{U}) = \mathbf{S}_b(\mathbf{U}) + \mathbf{S}_f(\mathbf{U}) + \mathbf{R}, \qquad (15)$$
>
> where $\partial_t$, $\partial_x$ and $\partial_t$ denote partial derivatives in the horizontal spatial dimensions $x$ and $y$, and temporal dimension $t$.

"Line 80: define L and T.
Line 83: I don't like calling Cf a coefficient since it includes a dynamic variable, i.e. h; rephrase; it is a friction function. I would take out h^1/3 (of Cf) and even write it is h^alpha since alpha is a parameterization anyway."
L and T were defined in the revised paper, and $C_f$ is referred to as "friction function". Apart from these requested revisions, the friction source term is written as it appears in Liang and Marche, 2009.

**2.1 The new LISFLOOD-DG2 solver**

Units are notated in square brackets [·], where L denotes unit length and T denotes unit time. The two-dimensional topographic elevation data is denoted $z$ [L] and $g$ is the gravitational acceleration [L/T$^2$]. The frictional forces in the $x$- and $y$-directions are $S_{fx} = -C_f u\sqrt{u^2 + v^2}$ and $S_{fy} = -C_f v\sqrt{u^2 + v^2}$ , where the  friction function is $C_f = gn_M^2/h^{1/3}$ and $n_M(x, y)$ is Manning's coefficient [T/L$^{1/3}$].

"Line 90: Why is this a dot product? No dot; it is a matrix-vector product."
This typo was removed.

"Line 119: there should be a flag doing that (application of a slope limiter) automatically for smooth solutions, not a hand-switch. See Krivodonova or extensions for improvements. In any real flooding case, this should be done via an automatic switch. Please clarify and add a sensible automatised switch. That the test cases do not involve shock wave propagation, is that not an omission? Tests 2 and 3 must have seen some shock-wave propagation prior to settling into a quasi-steady state, given the $F > 1$, $F < 1$ transition observed and discussed. Please clarify"
The local limiter is actually automatic, when activated, but can still entail around double the runtime costs as it needs to search where to apply or not apply slope limiting (Ayog et al. 2021). Informed by the study Ayog et al. (2021), flooding cases over a highly rough and dry topography do not need local slope limiting, and switching it off completely greatly accelerates DG2 solver's runtimes.

**2.1 The new LISFLOOD-DG2 solver**

While LISFLOOD-DG2 is equipped with a generalised minmod slope limiter (Cockburn and Shu, 2001) localised by the Krivodonova shock detector (Krivodonova 2004), the automatic local slope limiter was deactivated for the sake of efficiency: none of the test cases presented in Sect. 3  involve shock wave propagation   since all waves propagate over an initially dry bed and are rapidly retarded by frictional forces (Néelz and Pender, 2013; Xia et al., 2019). The lack of shock wave propagation means that all LISFLOOD-FP solvers—DG2, FV1 and ACC—are capable of realistically simulating all test cases presented in Sect. 3.

"Line 121: In KS2020, I do not see thus split implicit scheme explained? Rather KS2020 refers to KL2010, where matters are also not explained, so reference is inappropriate. LM2009 do show a split scheme but it is not really implicit. I find (8) to (9) vague. Too convoluted. Simply give the entire scheme in 9 (there is ample space) then we all know what is done rather then having to piece somehow together what is factually done (which I failed to do). Comment whether the schemes for Sf and rain are 2nd or 1st order also per the comment on

eqn (8) below. Given that Sf is kind of quadratic in u (or hu) a semi-implicit scheme is almost instantly made up. That would be straightforward for (9a) and even easy for (9b). One can check the formal time accuracy for the case with only Sf and R on the RHS. And perhaps even for a constant bottom slope river kinematic limit (which is an exact limit within the SWE)."

As mentioned above in our reply to major remark (i), subsection 2.1.2 has been revised to explain how the split friction discretisation of Liang and Marche, 2009 was integrated in our DG2, and in what sense is implicit. In addition, a new subsection 2.1.3 was added to explain how the rain term was integrated.

**2.1.3 Discretisation of the rainfall source term**

The discretisation of rainfall source term  evolves the water depth element-average coefficients $h_{i,j,0}$:

$$h_{i,j,0}^{n+1} = h_{i,j,0}^n + \Delta t R_{i,j}^n, \tag{14}$$

where $R_{i,j}^n$ denotes the prescribed rainfall rate at element $(i,j)$  and time level $n$. Eqn. (14) is first-order-accurate in space and time, which is deemed sufficient since rainfall data is typically available at far coarser spatial and temporal resolutions than the computation grid, leading to zero element-wise slope coefficients for the rainfall source ter~~min order to preserve the existing local water surface gradient. After applying friction and rainfall source terms, flow coefficients $\mathbf{U}_{i,j}$ are evolved from time level $n$ to $n+1$ using an explicit two-stage Runge-Kutta scheme (Kesserwani et al., 2010): where element indices $(i,j)$ are omitted for clarity of presentation. The time-step $\Delta t$ is calculated according to the CFL condition using the maximum stable Courant number of 0.33 (Cockburn and Shu, 2001). The~~. Recall that the rainfall source term, friction source term, and remaining flux and bed slope terms are treated separately such that, at each timestep, the flow variables updated by Eqn. (14) are subsequently updated by Eqn. (12), and finally by Eqns. (8)–(9). The complete DG2 model workflow is summarised by the flowchart in Fig. 2, wherein each operation is parallelised using the CPU and GPU parallelisation strategies discussed next.

"Line 121: Why is DG2 slower; due to the CFL=1/3 criterion relative to CFL $\sim$ 0.9 for FV1? Eqn (8): why is this scheme for rainfall 2nd order? Why is rain not directly included into (9). R=R(t) so non-autonomous RK would be fine? Please explain your choices better. Please provide the entire scheme for the entire system in (9); there is space and currently the time stepping scheme is unclear. Please clarify." Specification for the CFL number with DG2 and FV1:

> ### 2.1 The new LISFLOOD-DG2 solver
>
> The remaining terms are the spatial fluxes and topographic slope terms, which are discretised by an explicit second-order two-stage Runge-Kutta scheme (Kesserwani et al., 2010) to evolve the flow coefficients  $\mathbf{U}_{i,j}$ from time level $n$ to $n+1$:
>
> $$\mathbf{U}^{\text{int}} = \mathbf{U}^n + \Delta t \, \mathbf{L}(\mathbf{U}^n), \tag{15a}$$
>
> $$\mathbf{U}^{n+1} = \frac{1}{2} \left[ \mathbf{U}^n + \mathbf{U}^{\text{int}} + \Delta t \mathbf{L}(\mathbf{U}^{\text{int}}) \right], \tag{15b}$$
>
> where element indices $(i,j)$ are omitted for clarity of presentation. The initial time-step $\Delta t$ is a fixed value specified by the user, and the time-step is updated thereafter according to the CFL condition using the maximum stable Courant number of 0.33 (Cockburn and Shu, 2001).

Clarification on why the rain terms is not directly integrated in equation 9:

> ### 2.1 The new LISFLOOD-DG2 solver
>
> A standard splitting approach is adopted such that the friction source term $\mathbf{S}_f$ and rainfall source term $\mathbf{R}$ in Eqn. (15) are applied separately at the beginning of each time-step. By adopting a splitting approach, friction or rainfall source terms are only applied as required by the particular test case, for better runtime efficiency. The discretisation of the friction source term is described later in Sect. 2.1.2, and the rainfall source term in Sect. 2.1.3.

"Figure 2: How can one apply rainfall if dt is not know yet, since dt is determined later in the diagram? Order seems off?"
The revised manuscript clarifies that, "The initial time-step is a fixed value specified by the user and the time-step is updated thereafter according to the CFL condition". Figure 2 has also been clarified by using 'Update $\Delta t$' instead of 'Calculate $\Delta t$'.

"Line 157: typo; L missing (only subscript y seen)." "Line 173: add OUP 'and,' at end of third option.
Typo fixed and an 'and' was added.

"Line 183: Can it be clarified earlier whether or not (I think it is "or not") DG2 can be truly Cˆ0? Since the x*y term is not include, therefore it cannot represent Cˆ0 solutions or topography. While speed is gained this drops formal Cˆ0 smoothness. Say so explicitly, probably earlier, also why this option with only polynomials 1, x and y is chosen, I assume to enhance computational speed."
The introductory text added at the start of subsection 2.1 summarises the rationale for adopting the proposed DG2 formulation. Also, it is stated in the added subsection 2.1.1 the continuity property for the DG2 topography projection is only satisfied at the face-centers where wetting-and-drying treatment and Riemann fluxes are evaluated (i.e. at the location

that matters the most!). A theoretical proof can be found in (Kesserwani et al., 2018). (Revised text shown in reply to an earlier comment.)

"Line 186: Nothing is said about the time steps for FV1 and ACC? Why does FV1 do better for thin overland flows? How are Sf and R dealt with in FV1? This information is relevant to understand the test results provided and analysed later. Please clarify (earlier on)."
The CFL numbers used with FV1 and ACC were added in Section 2; and, the added subsection 2.1.2 offers an explanation on why FV1 can better handle thin overland flows.

"Eqn (14): Perhaps add some spacing around $\Delta t$."
Space added.

"Line 207: Second-order wrt to what since it cannot be 2nd order wrt solutions requiring the advective velocity terms? Please define clearly what is meant here? When advection is important, it cannot be second order, of course?"
Clarification was made to explain in what sense ACC offers a second-order discretisation in space (see also the revised Appendix B). As the set of PDEs used for the ACC solver neglect the advective velocity terms, we agree that its discretization cannot be valid for advection dominated flow, irrespective of the order-of-accuracy of its spatial discretisation.

"Fig. 6: Since FV1 is order one that use of the same grid spacing in the different models is a priori unfair. What happens if FV1 is adjusted (and maybe ACC) such that the expected errors are the same? Please add that situation. The left bottom panel could be done for finer resolution as well, as in Fig 7"
Answered in our reply to major remark (iv).

"Fig. 7: Figure a bit large. Otherwise nice"
The figure size is based on the line length, which is full-width in the proof version, but the figure will only occupy a single column in the final publication.

"Line 270 paragraph: But this comparison is unfair; one needs to compare run times for the same or at least similar accuracy; either focussing on overall accuracy or the accuracy at the front since the latter may be most relevant. This needs an extension.
Line 283: Statement by authors themselves underscores my previous remark. So why is this case not made in the paper with the same convergence/accuracy compared with runtime?
Fig. 8b,c: Same remark; what is the right comparison here? Please address. Caption: page number and caption overlap."
Answered in our reply to major remark (iv).

"Fig. 12: Can you show simulations for DG2 with dx=50m, say, as well since that runs in less time than the other models but the solution may still be better?
Table 1: So DG2 dx=50m should be compared with runtimes of the other models or even DG2-GPU at dx=25m should be compared with the other models, in line with the previous five remarks."
Addressed in our reply to major remark (iv).

"Line 352: Why is the domain initially dry? How long does it take for the memory of this odd initial condition to disappear?"

This model spin-up procedure is originally specified by Xia et al., 2019, and the memory of the initially dry condition disappears after 1.5 simulated days.
* * *
**3.3 Catchment-scale rain-on-grid simulation**

As specified by Xia et al., 2019,  the simulation comprises a spin-up phase and subsequent analysis phase. The spin-up phase starts at 00:00 3 December 2015  from an initially dry domain. Water is introduced into the domain via the rainfall source term (Eqn. 14), using Met Office rainfall radar data at a 1 km resolution updated every 5 minutes (Met Office, 2013).  The spin-up phase ends and the analysis phase begins at 12:00 4 December 2015,  once the memory of the dry initial condition has disappeared, and water depths and discharges in all river channels have reached a physically realistic initial state (Xia et al., 2019). The simulation ends at 12:00 8 December 2015 after a total of 5.5 simulated days.
* * *
"Line 356: I do not understand this remark 'No Data'; translate this computer remark."
The explanation has been revised to avoid using computer jargon:
* * *
**3.3 Catchment-scale rain-on-grid simulation**

An open boundary condition is imposed along the irregular-shaped catchment perimeter by adjusting the  terrain elevation of elements lying outside the catchment such that their elevation is below mean sea level, thereby allowing water to drain out of the River Eden into the Solway Firth. At each time-step, water flowing  out of the Solway Firth is removed by zeroing the water depth in elements lying outside the catchment.
* * *
"Line 364: But that seems unfair. Finer data need to be projected on the topography that DG2 can model, which is not dx=40m but the finer one projected? In fact, DG2 would then do a bit better I suspect. Address this please; perhaps really redo the DG2 case; it should have 3 dofs per cell, also for the topography."
Answered in our reply to major remark (ii).

"Line 367: Why is repositioning required since across the river channel the river level is within 0.2m, say, the same." Approximate gauging station coordinates are provided by the UK Environment Agency, but these are often positioned near the river bank and not in the channel itself.
"Is the level meant the river level height above sea level?" Correct. For each model run, hydrographs of free-surface elevation above mean sea level are measured in river channels at sixteen gauging stations.
"What is recorded?" Observed free-surface elevation hydrographs are calculated from Environment Agency measurements of water depth and river bed elevation above mean sea level.

"So why is this repositioning needed, since variation across the river channel will be minimal." Approximate gauging station coordinates are provided by Environment Agency, 2020, but these are often positioned near the river bank and not in the channel itself.

"It is free-surface height that matters. Please clarify." Yes, this is a more specific term, so we now use "free-surface elevation" consistently across Section 3.3.

"Fig. 14: I am not convinced by the DG2 results due to not taking into account slope information in the bathymetry. So DG2 may be underperforming while this can partially be repaired. Please update and clarify. Line 408: As stated above this anomaly of using only a planar representation in DG2 should be fixed: you do not use DG2 optimally with only 40m planar resolution on bathymetry, while you are allowed to include higher resolution and project on the DG2 basis you use. This leads to an unfair comparison which seems not aligned with the DG2 capabilities."

Answered in our reply to major remark (ii).

"Line 418: What is the error in the observations? What is the variation in surface level across a (flooded) river channel? What is consequently the result of shifting the measurement position? I would guess the error is circa 0.2m (from watching river levels bob about ±0.1m for a river in flood next to a river gauge). What is the variation within 10m along stream of the surface level in the simulation and across the channel, in order to get an error estimate? Please clarify and add a discussion on errors, in both the observations and simulations."

As these high-frequency temporal variations average out in our long-running simulation, we wouldn't count them as observation errors. A discussion on the expected observation errors has been added:

**3.3.1 River channel free-surface elevation hydrographs**

 Free-surface elevation hydrographs at the sixteen river gauging stations are shown in Fig. 14. Observed free-surface elevation hydrographs are calculated from Environment Agency measurements of water depth and river bed elevation above mean sea level (Environment Agency, 2020). While water depths can be measured to an accuracy of $\sim 0.01$ m (Bates et al., 2014), discrepancies between in-situ, point-wise river bed elevation measurements and the remotely-sensed, two-dimensional DEM can result in systematically biased free-surface elevations, as reported by Xia et al., 2019.

"Line 425: Please add the word 'maximum' before the last word 'flood'."
Added.

"Line 428: 'More notable differences' Where? Clarify please, e.g. by using arrows. Fig. 16: Graphs in Fig. 16 look all the same; use arrows to highlight areas you want the reader to focus on."
Arrows were added, see revised Fig. 16.

"Line 428: On flood defence walls not being captured on coarse grids: unless, of course, you put those in by hand, which should be done (regardless). Here is where variable/non-uniform mesh capabilities of DG2 or FV1 would have come in handy. These are then essentially

(vertical) walls with a finite vertical extent. Also on a coarse mesh one can add vertical walls at cell edges as approximation"
Answered in our reply to major remark (iii).

"Line 432: So, what then is right or wrong, ACC or FV1 and how do we know that? Please clarify"
The text has been revised to clarify that ACC at the finest resolution, of 10m, was used as the reference solution and appropriate metrics to measure floodplain prediction capability were used.

> ### 3.3.2 Maximum flood extent over Carlisle
>
> The DG2 and FV1 predictions of maximum flood extent can be quantified against the ACC prediction at $\Delta x = 10$ m, which is treated as the reference solution. The hit rate measures flood extent underprediction as the proportion of wet elements in the reference solution that were also predicted as wet. The false alarm ratio measures flood extent overprediction as the proportion of predicted wet elements that were dry in the reference solution. The critical success index measures both over- and underprediction. All three metrics range between 0 and 1, and further details are provided by Wing et al., 2017.

"Fig. 17: I still disagree with a DG0 bottom topography at dx=40 for DG2. That is a false comparison, cf. previous remarks made."
Answered in our reply to major remark (ii).

"Further: what flood accuracy do we actually care about and what are the error bars given errors in rainfall? Please discuss." The following text has been added to clarify the level of accuracy for the considered case study.

> ### 3.3 Catchment-scale rain-on-grid simulation
>
> While rainfall data errors can influence model outputs, (Ming et al., 2020) found that a prescribed 10% rainfall error lead to only 5% relative mean error in predicted water depth hydrographs. As such, modelling uncertainties due to rainfall errors are not quantified in these deterministic model runs.

"Line 491: But as said before, your topography is incorrect, in not matching DG2 accuracy that can be used. Also, I am not convinced by your time stepping in DG2 as compared to what is done in ACC and FV1 (the latter which is unclear as well and makes it difficult to understand the comparison). Please clarify"
Answered in our reply to major remark (ii).

"Line 501: The conclusion here is a bit of a downer for DG2; no discussion is made on using hybrid approaches, including variables time and spatial resolution for which DG should be good and more promising. Please comment and update."
Answered in our reply to major remark (vi).

"Appendix B: I am a bit confused; why do we need to linearise to establish whether ACC is second order? Please clarify." Linearisation is necessary in order to perform a Taylor series expansion of the discrete local inertia equations. The manuscript now justifies this linearisation more clearly:

**Appendix B: LISFLOOD-ACC order-of-accuracy**

The formal order-of-accuracy of LISFLOOD-ACC is determined by  a numerical analysis of the discrete local inertia equations (de Almeida et al., 2012).  To begin, the local inertia equations  are linearised by assuming small perturbations in free-surface elevation $\eta$ about a constant reference depth $H$ [L], leading to the linearised frictionless one-dimensional local inertia equations:

$$\frac{\partial \eta}{\partial t} + \frac{\partial q}{\partial x} = 0, \tag{B1a}$$

$$\frac{\partial q}{\partial t} + gH\frac{\partial \eta}{\partial x} = 0, \tag{B1b}$$

This linear assumption is valid for gradually-varying, quasi-steady flows (de Almeida et al., 2012), and ensures that the remainder of the analysis is tractable.